# Near-infrared manipulation of multiple neuronal populations via trichromatic upconversion

Xuan Liu[1], Heming Chen[1], Yiting Wang[1], Yueguang Si[1], Hongxin Zhang[1], Xiaomin Li [1✉], Zhengcheng Zhang[1], Biao Yan [1], Su Jiang[2], Fei Wang[2], Shijun Weng[1], Wendong Xu[1,2], Dongyuan Zhao [1], Jiayi Zhang [1✉] & Fan Zhang [1✉]

Using multi-color visible lights for independent optogenetic manipulation of multiple neuronal populations offers the ability for sophisticated brain functions and behavior dissection. To mitigate invasive fiber insertion, infrared light excitable upconversion nanoparticles (UCNPs) with deep tissue penetration have been implemented in optogenetics. However, due to the chromatic crosstalk induced by the multiple emission peaks, conventional UCNPs or their mixture cannot independently activate multiple targeted neuronal populations. Here, we report NIR multi-color optogenetics by the well-designed trichromatic UCNPs with excitation-specific luminescence. The blue, green and red color emissions can be separately tuned by switching excitation wavelength to match respective spectral profiles of optogenetic proteins ChR2, C1V1 and ChrimsonR, which enables selective activation of three distinct neuronal populations. Such stimulation with tunable intensity can not only activate distinct neuronal populations selectively, but also achieve transcranial selective modulation of the motion behavior of awake-mice, which opens up a possibility of multi-color upconversion optogenetics.

[1] Department of Chemistry, Institutes of Brain Science, State Key Laboratory of Molecular Engineering of Polymers, State Key Laboratory of Medical Neurobiology, MOE Frontiers Center for Brain Science, Fudan University, Shanghai 200433, P. R. China. [2] Department of Hand Surgery, Huashan Hospital, Priority Among Priorities of Shanghai Municipal Clinical Medicine Center, National Clinical Research Center for Aging and Medicine, Department of Hand and Upper Extremity Surgery, Jing'an District Central Hospital of Shanghai, Shanghai 200040, China. ✉email: lixm@fudan.edu.cn; jiayizhang@fudan.edu.cn; zhang_fan@fudan.edu.cn

Precise modulation of brain activity has long been desired for brain function dissection, neurologic disease treatment[1,2], and brain–computer interfaces[3,4]. Optogenetics is proven to be widely applied in neuroscience to dissect the roles of one specific type of neurons in numerous neural circuits that controls sensory, motor, memory, emotion, and social behaviors. Information processing in cortical and other circuits requires integrating inputs over a wide range of spatial scales, from local microcircuits to long-range connections. Dissecting the function of each neuronal type may not require simultaneous activation, but a quantitative comparison of the strengths of multiple pathways needs cell-type-specific manipulation in multiple neuronal populations. Therefore, investigating multiple pathways is desired but remains to be a big challenge, which requires coordinated activation and suppression of different neuronal types in the same brain area in one preparation. Independent control of different cell types is workable by multicolor modulation[5–8]. Thanks to the establishment of optogenetic protein family[9] as well as the genetic tools to target various cell types in the nervous system[10], we are now able to develop a toolkit for multicolor optogenetics.

The potential clinical applications of optogenetics include epileptic suppression and pain relief[11,12]. Due to the presence of skull and skin, as well as the short penetration depth of visible light, optogenetic stimulation normally requires the invasive insertion of optical devices[13–15]. Near-infrared (NIR) excited upconversion nanoparticles (UCNPs)[16] with narrow emission bandwidths and anti-photo-bleaching show advantages of deep tissue penetration in comparison with visible lights[17], which have been utilized for noninvasive remote activation of neurons[18–20] in zebrafish larvae and mice, as well as an extension of NIR vision[21].

The major problem of conventional UCNPs used in multicolor optogenetics is that single NIR excitation generates coexisting multiple emission bands from different types of monochromatic UCNP. Directly mixing monochromatic UCNPs leads to cross talk between different neuronal populations that cannot be independently controlled in a multicomponent mixture. Independent activation of distinct neural populations requires excitation-specific non-cross-talking multicolor emissions and continuous wave (CW) light with pulse modulation for temporal control of spiking as well. But the excitation-responsive upconversion luminescence is still limited to only two colors as green and blue emission under CW NIR excitations[22]. Although distinct trichromatic color emissions have been realized via the nonsteady state upconversion[23], the requisite of specific laser pulse mode would restrict the modulation pattern when applied in complex neuronal stimulation.

Here we report a CW NIR excitable trichromatic UCNPs for remote optogenetic modulation of three distinct neuronal populations (Fig. 1a). For the core-multishell structured trichromatic luminescent UCNPs, blue, green, and red color emissions are triggered respectively with corresponding 980, 808, and 1532 nm CW lasers (Fig. 1b and Supplementary Movie 1). In contrast, the mixture of three different monochromatic UCNPs shows multicolor emissions (Fig. 1b). As a proof-of-concept, we use the trichromatic UCNPs to selectively activate three types of genetically targeted neurons, namely ChR2-expressing inhibitory parvalbumin (PV), ChrimsonR-expressing somatostatin (SOM) neurons, as well as C1V1-expressing excitatory $Ca^{2+}$/calmodulin-dependent protein kinase II α (CaMKIIα) neurons in the primary visual cortex (V1) (Fig. 1a). Three different NIR lights, can then penetrate into the brain tissue and excite the trichromatic UCNPs to activate PV, SOM, and CamKIIα neurons, respectively.

## Results

### Trichromatic UCNPs with NIR excitation-specific luminescence.
We designed the delicate multilayer core-shell nanostructure

$NaErF_4@NaYF_4@NaGdF_4{:}20\%Yb,2\%Er@NaGdF_4{:}20\%Yb@NaGdF_4{:}$
$50\%Nd,10\%Yb@NaYF_4@NaGdF_4{:}80\%Yb,1\%Tm@NaYF_4$ to realize trichromatic emissions via precise control the orientation of excitation energy[24] (Fig. 2a and Supplementary Fig. 1). The core-multishell structure can be divided into three parts by $NaYF_4$ inert layers. Each part performs a relatively independent upconversion process to realize trichromatic emissions: red emission peaked at ~650 under 1532 nm excitation, green emission peaked at ~540 under 808 nm excitation, and blue emission around 450/475 nm under 980 nm excitation. This trichromatic luminescence property maintains well at particle scale, which can be proved by spectrograph-coupled optical microscopy under the excitation of tightly focused CW lasers[25,26] of 1532, 980, and 808 nm respectively (Fig. 2b, c and Supplementary Fig. 2). The trichromatic optical property further allows gradient manipulation of multicolor emission by manipulating the excitation sources. Three monochromatic channels can be individually altered and thus variable visible color on demand can be regulated by synchronously tuning the power density of CW excitations (Fig. 2d and Supplementary Table. 1). The full-color emission can be locally generated and dynamically tuned (Supplementary Fig. 3). Owing to the well isolation of emission parts, each primary color performs preferable monochromaticity and thus the luminescence can cover a wider color gamut than previous reports (Fig. 2e). In addition, the trichromatic emission is power-dependent and also performs long-term non-blinking and non-photobleaching features (Supplementary Figs. 4, 5).

### Principle of regulating excitation energy for excitation-specific multicolor emission from the core-shell structured UCNPs.
Regulating the incident excitation energy to specific activators in the separate layers of the core-multi-shell nanostructure is crucial for realizing trichromatic luminescent property[27]. In our design, for integrating 1532 nm excited primary red emission from the erbium activators in the innermost core, the most challenging problem lies in blocking of 980 and 800 nm excitation energy between different sensitizers. We take the $NaErF_4@NaYF_4@$-$NaGdF_4{:}x\%Yb/y\%Tm@NaYF_4$ core-shell UCNPs as a typical example to investigate the dissipation effect of 980 nm first. Since both $NaErF_4$ core and $NaGdF_4{:}Yb/Tm$ shell can absorb 980 nm photon to generate red and blue upconversion emissions at the same time, the suppression of red emission from $NaErF_4$ core is crucial for the realization of primary blue emission from $NaGdF_4{:}Yb/Tm$ shell, which depends on the 980 nm excitation energy capturing capability of it. It has been demonstrated that 980 nm photons can be absorbed by $Yb^{3+}$ sensitizer. However, $NaGdF_4{:}10\%Yb$ shell with solely sensitizers can only induce less than 20.0% red emission suppression of $NaErF_4$ core (Fig. 3a). An increase of $Yb^{3+}$ to 49% would enhance the suppression to ~26.0%, but it is limited to only ~31.0% suppression even when $Yb^{3+}$ doping concentration is increased up to 80% (Fig. 3a). Therefore, simply increasing $Yb^{3+}$ sensitizers in the shell brings little effect to dissipate 980 nm excitation energy. Instead of sensitizer absorption effect, in the present work, we discover that even only 1% of $Tm^{3+}$ activator co-doped with 49% $Yb^{3+}$ sensitizer results in highly enhanced red emission suppression of inner core (~56.4%, Fig. 3b) while the red intensity remains almost the same under 1532 nm excitation or 808 nm (Fig. 3c and Supplementary Fig. 6), which indicates a selective dissipation of 980 nm excitation energy. The lifetime of $Yb^{3+}$ sensitizer under 980 nm excitation shows a great decrease from 828 to 101 μs when adding and increasing $Tm^{3+}$ activators from 1 to 10% (Fig. 3d). This suggests that activators ($Tm^{3+}$) may lower the active state population density of sensitizers ($Yb^{3+}$) and decrease the lifetime, which allows a higher population of excitation energy extraction from sensitizer to an activator in the shell (Supplementary Fig. 6). In

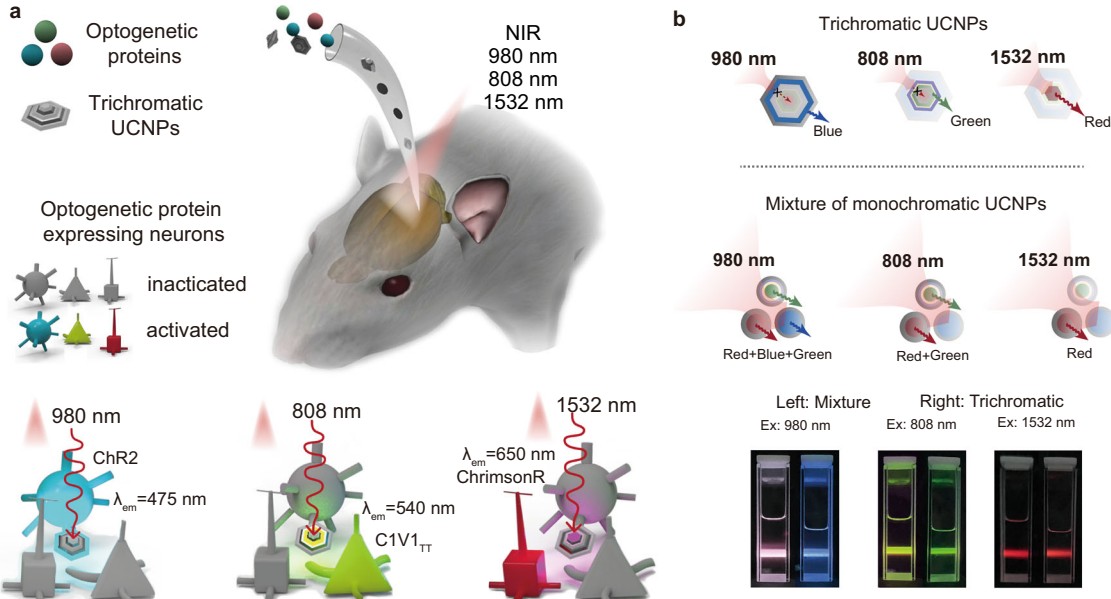

**Fig. 1 Modulation of multiple neuronal population with trichromatic UCNPs. a** Schematic showing NIR trichromatic upconversion mediated multicolor optogenetics. Channelrhodopsins and trichromatic UCNPs were introduced into specific brain areas. 980, 808, and 1532 nm NIR light can penetrate brain tissue and excite the trichromatic UCNPs respectively, which would emit corresponding 475, 540, and 650 nm visible color and activate the matching neuronal population. **b** Schematic showing the CW NIR excited multicolor emission of trichromatic UCNPs and the mixture of monochromatic UCNPs with correspondent photographs under three NIR excitation. NIR laser of 980, 808, or 1532 nm only generates one corresponding color in trichromatic UCNPs with dissipation core-shell structure, namely blue, green, and red emission. A mixture of conventional monochromatic UCNPs under NIR excitation. 808 and 980 nm NIR light induce two or three color bands emission.

comparison, under 980 nm excitation, a lifetime of 1532 nm emission from the inner $NaErF_4$ core remains unaltered (Supplementary Fig. 6). Nevertheless, further increase of $Tm^{3+}$ concentration from 1 to 10% contributes little enhancement of 980 nm dissipation but severe quenching of blue emissionintensity by cross-relaxation (CR, Fig. 3b). Therefore doping concentration of the $Tm^{3+}$ activator is chosen to be 1% in the subsequent multicolor construction. To maximize the excitation energy dissipation, $Yb^{3+}$ should be increased, but over 80% of $Yb^{3+}$ shows a shrinking of particle size and decrease of $Tm^{3+}$ emission intensity on the contrary (Supplementary Fig. 7). So that we chose 80% of $Yb^{3+}$ to enhance the energy transfer (ET) and finally realizing 64.4% red emission suppression of the inner core while keeping the intense $Tm^{3+}$ blue emission as well (Fig. 3e and Supplementary Fig. 8). These results suggest that $NaGdF_4$:Yb/Tm shell can be used to dissipate the excitation energy, the mechanism follows two steps: First, the activator favors the excitation energy extraction via ET from sensitizer. Then the subsequent radiated emission of activators and CR among dopants dissipate such energy. This process dominates the excitation energy dissipation (Fig. 3f). So that increasing $Tm^{3+}$ and $Yb^{3+}$ doping enhances dissipation efficiency by inducing more ET and CR.

Besides, the thickness of the shell has an obvious effect on excitation energy dissipation. Red emission from the inner core can be gradually suppressed by increasing shell thickness from ~4 to ~11 nm (Supplementary Fig. 9). Meanwhile, with the same shell thickness, excitation energy dissipation is always stronger in the presence of the $Tm^{3+}$ activator compared with the $Yb^{3+}$ sensitizer only. Yet changing the thickness alone does not have a significant effect on the lifetime of sensitizers in a shell.

Except from the 980 nm dissipation process which is related to sensitizer $Yb^{3+}$ and $Er^{3+}$, the other two cases of dissipation are involved in the trichromatic emission as well (Supplementary Fig. 10). The dissipation of 808 nm is related to the sensitizer of $Nd^{3+}$ and $Er^{3+}$, which shows similar results (Supplementary

Fig. 11). Similarly, the case of dissipation between the same type of sensitizers ($Yb^{3+}$ or $Nd^{3+}$) in both core and shells is investigated (Supplementary Fig. 12).

Finally, the $NaErF_4@NaYF_4$ core are coated with over ~10 nm $NaGdF_4$:Yb/Tm layer to realize the orthogonal red and blue upconversion luminescence under the excitation of 1530 and 980 nm, respectively (Supplementary Figs. 13, 14).

**Activation of distinct neuronal populations independently under different NIR excitations with trichromatic UCNPs.** Blue, green, and red emissions from our trichromatic UCNPs can be separately excited with 980, 808, and 1532 nm light and match well with the reported activation spectra of three optogenetic proteins: ChR2, C1V1, and ChrimsonR[5,28] (Figs. 1a, 4a). In contrast, directly mixing three types of monochromatically emissive UCNPs with blue, green, and red colors ($NaGdF_4$:80% Yb,1%Tm@$NaYF_4$–blue, $NaGdF_4$:20%Yb,2%Er@$NaYF_4$-green, and $NaErF_4@NaYF_4$-red) resulted in three- or two-color emission peaks under single NIR excitation at 980 or 808 nm, making it difficult to activate multiple optogenetic proteins independently (Fig. 4a). To show the utility of our trichromatic UCNPs in in vivo optogenetic experiment, a mixture of AAV-DIO-ChR2-mCherry, AAV-CaMKIIα-C1V1-EYFP, and AAV-fDIO-ChrimsonR-mCherry viruses was stereotaxically injected into the visual cortex (V1) of SOM-Flp: PV-Cre mice (Fig. 4b). ChR2, C1V1, and ChrimsonR were selectively expressed in PV, CaM-KIIα, and SOM neurons, respectively (Fig. 4c). Trichromatic UCNPs dissolved in saline was then injected into the same region 2 weeks after the viral injection (Fig. 4d). No significant aggregation of astrocyte was detected while there was a small amount of microglia aggregation around the injection site up to 2 weeks after UCNPs injection (Supplementary Fig. 15). The distribution of UCNPs was observed by HRTEM, suggesting the majority were distributed in extracellular space around the injection site and a small fraction of which may be uptaken by microglia[18]

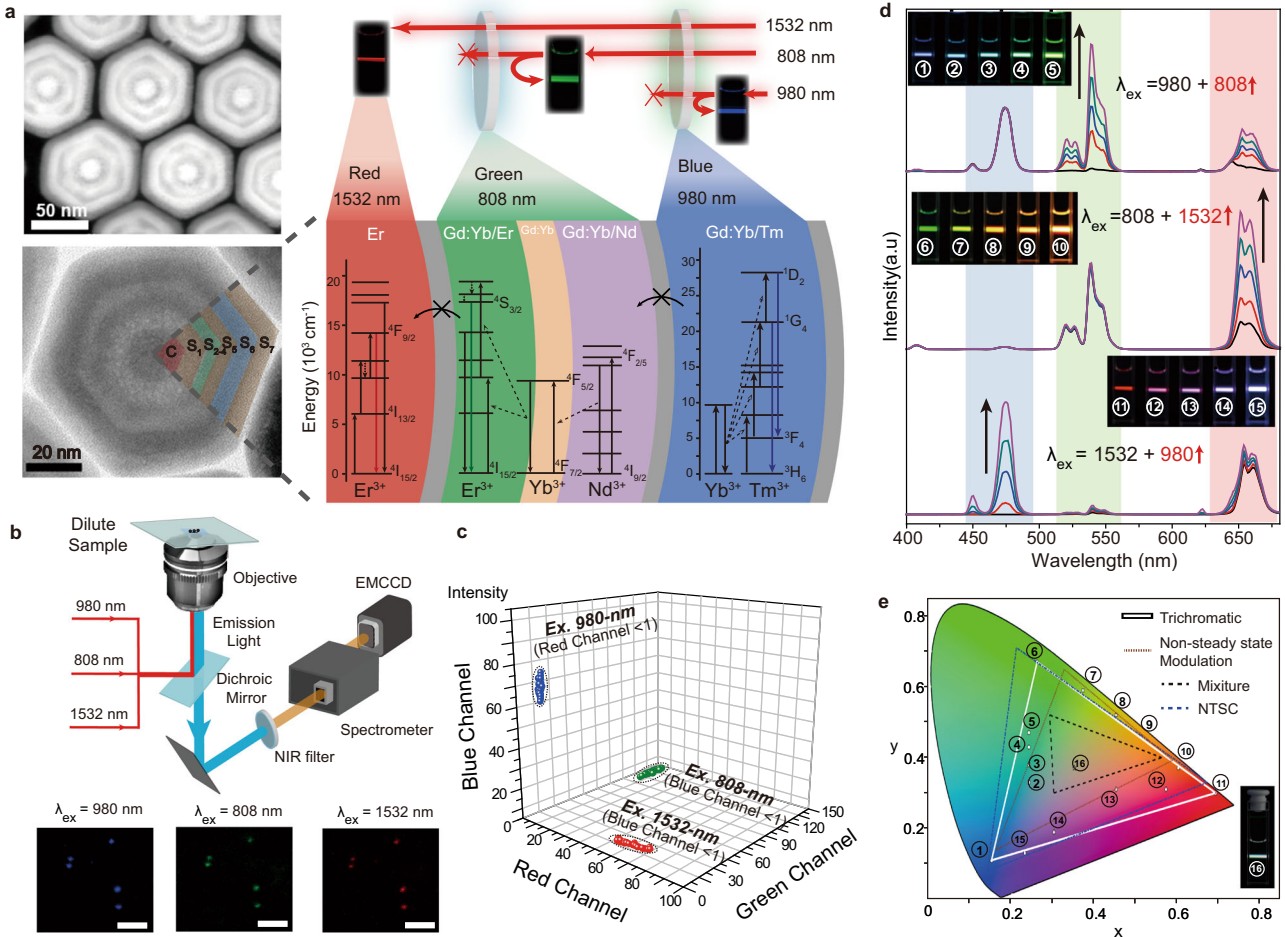

**Fig. 2 Optical property of trichromatic UCNPs. a** The high-angle annular dark field scanning transmission electron microscopy image (top-left) and high-resolution transmission electron microscopy (TEM) image (bottom-left) of the obtained multilayer UCNPs and the corresponding schematic illustration of energy dissipation upconversion process in multilayer UCNPs. The synthesis was repeated four times with similar results. **b** Spectrograph-coupled optical microscopy setup for single-particle imaging and in situ spectral analysis and upconversion luminescent images of individual UCNPs spots under 808, 1532, and 980 nm excitation. Scale bar, 3 μm. The imaging test was repeated by twice with individual samples. **c** Scattering plots of RGB channel intensity of single-particle luminescent spots under different excitations. Images and corresponding spectra were collected under ~120 mW/mm² excitation with 60 s exposure time. RGB channel intensity was calculated from average intensity within a specific wavelength range (438–480 nm for blue, 507–560 nm for green, 640–700 nm for red). **d** Upconversion luminescence spectra and corresponding photographs of trichromatic UCNPs under dynamic tuning of dual-excitation laser power. **e** Color gamut of the tuning emissions point (1) to (16) in **d** (marked with the white triangle), nonsteady state modulation (marked with the brown dotted triangle), and the National Television Standards Committee (NTSC) standard (marked with the black dotted triangle). Insert: white color at point (16) under co-irradiation of three lasers. The corresponding color space area reported in this work cover 84% of NTSC. Source data are provided as a Source Data file.

(Supplementary Fig. 16). The UCNPs also showed stable photoluminescence output in the physiological environment for over a week (Supplementary Fig. 17). Tested by elemental analysis, the mass of UCNPs maintained about 77% after injecting into the brain for 3 weeks.

We first investigated the chromatic selectivity of the three optogenetic proteins[5]. In an in vitro brain slice preparation, the protocol describing electrophysiology recording via UCNPs mediation can be found at Protocol Exchange[29]. It is confirmed that 470 nm stimulation elicited more spikes in ChR2-expressing cells than C1V1(T/T)-expressing cells (activation ratio ~0.8:0.5) (Supplementary Fig. 18). Similarly, 546 nm light elicited more spikes in C1V1(T/T)-expressing cells than ChrimsonR-expressing cells (activation ratio ~0.8:0.4) (Supplementary Fig. 18). We then confirmed the selectivity under NIR illumination by showing that ChR2-expressing and C1V1(T/T)-expressing cells were selectively activated by 980 nm and 808 nm (Supplementary Fig. 19). In vivo

recordings of V1 neurons further showed that blue (470 nm), green (530 nm), and red (630 nm) visible lasers selectively activated ChR2-, C1V1-, and ChrimsonR-expressing neurons, respectively (Fig. 4a, e, Supplementary Table 2, and Supplementary Fig. 20).

The upconverted emissions power density at specific tissue depth was calculated by measuring upconverting efficiency and NIR tissue penetration ratio (Supplementary Figs. 21, 22). Subsequently, by adjusting NIR power density at the brain surface, it can be made sure that visible emission power density is no more than visible lasers power density at the specific depth in the brain, thus ensure the chromatic selectivity of the three optogenetic proteins.

The heating effect is crucial for the NIR remote modulation. Previous studies suggested that brain tissue would be irreversibly damaged when the temperature increases more than 6–8 °C[30,31]. The heat here is mainly generated by tissue absorption of NIR and

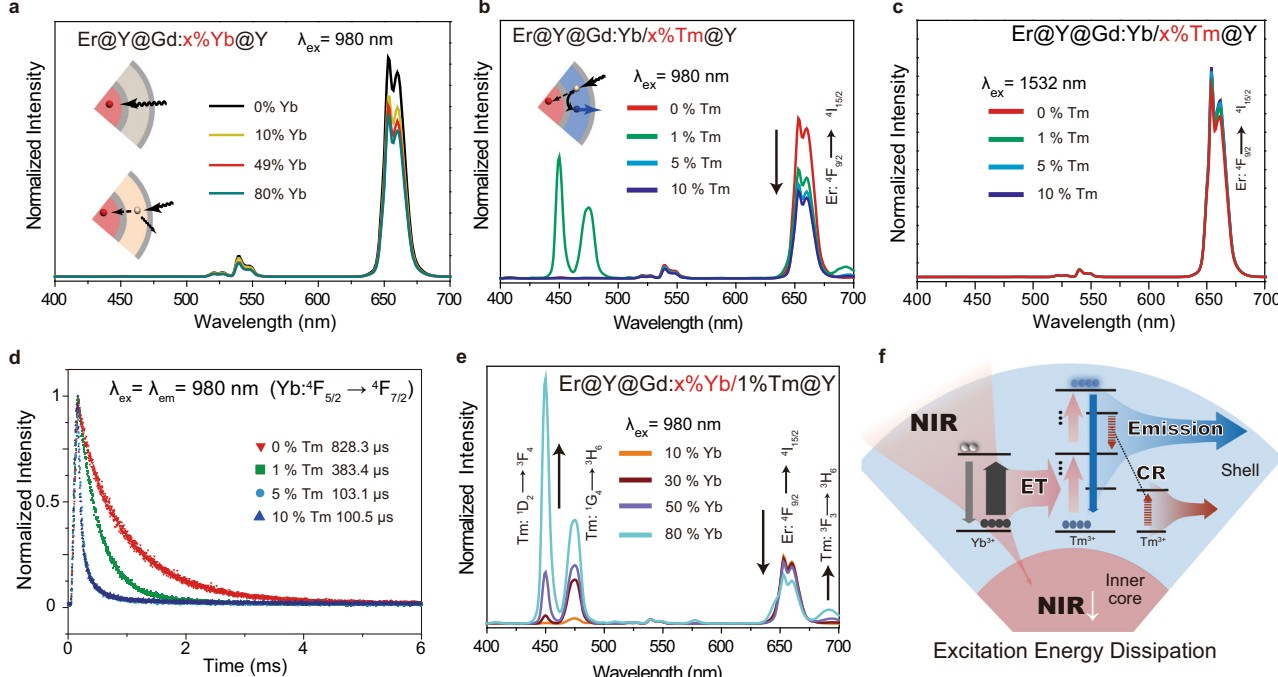

**Fig. 3 Investigation of energy dissipation based process. a** Emission spectra in the red channel of NaErF$_4$@NaYF$_4$@NaGdF$_4$:x%Yb@NaYF$_4$ UCNPs (x = 0, 10, 49, 80) with different shell composition. **b**, **c** Emissions spectra of NaErF$_4$@NaYF$_4$@NaGdF$_4$:49%Yb/y%Tm@NaYF$_4$ UCNPs (y = 0, 1, 5, 10). **d** Luminescence decay curves at 980 nm of the obtained UCNPs with different Tm$^{3+}$ doping ratios under 980 nm excitation. λ$_{em}$ emission wavelength. **e** Emissions spectra of NaErF$_4$@NaYF$_4$@NaGdF$_4$:x%Yb/1%Tm@NaYF$_4$ UCNPs (x = 10, 30, 50, 80). All of the UCNPs samples in these tests were dispersed in cyclohexane for the collection of spectra. Shell thickness is tuned around ~4 nm. λ$_{ex}$ excitation wavelength. Power density is set at 45 mW/mm$^2$. **f** Diagrams illustrating the energy dissipation dominating process. Suppression of the inner core's emission is substantially dominated by excitation energy dissipation, which can be enhanced through increasing energy transfer (ET) and subsequent energy consumption from emission or cross-relaxation (CR). Source data are provided as a Source Data file.

sharply decreases when penetrating brain tissue. So that heating effects were evaluated from the brain surface, which has the highest temperature during NIR light irradiation (Supplementary Fig. 23). To reduce the NIR heating effect, a laser pulse of 0.05 Hz was chosen here for in vivo electrophysiology recording (Supplementary Fig. 24). With the same NIR power density and intermittent irradiation condition used in in vivo electrophysiological experiments, it is found that the overall temperature on the brain surface under 980, 808, and 1532 nm illumination increased from 30.61 to 33.28 °C (Δ°C = 2.67 ± 0.18 °C), 30.62 to 33.44 °C (Δ°C = 2.82 ± 0.09 °C), and 30.78 to 33.99 °C(Δ°C = 3.21 ± 0.08 °C) (Supplementary Fig. 25a, b). The final NIR laser powers were chosen in the allowed range (Fig. 4f) that is between the response detectable threshold and the photothermal upper limit of NIR light (Supplementary Figs. 23–25).

Neurons exhibited distinct responses under selective activation with 980, 808, or 1532 nm NIR light (Fig. 4g). The protocol describing electrophysiology recording via UCNPs mediation can be found at Protocol Exchange[32]. We identified the types of neurons by analyzing the spike waveforms of all recorded neurons including experiments of visible lights and NIR light with trichromatic UCNPs. The majority of 530 and 808 nm-responsive neurons (CaMKIIα neurons) were regular-spiking cells with similar waveforms (putative pyramidal neurons), while both 470 and 980 nm-responsive neurons (PV neurons) were fast-spiking cells with similar waveforms (putative inhibitory interneurons). Similar to the reported results that SOM neurons showed variance in spike width[33], 650 and 1532 nm-responsive neurons (SOM neurons) had both fast-spiking and broad-spiking features (Supplementary Fig. 26). We also conducted control experiments to show that no significant responses were recorded without UCNPs or optogenetic protein expression

(Supplementary Fig. 27). Note that there was a relatively low undesired red emission under 808 nm stimulation. The emission power ratio of green/red was ~3:1, and the response ratio of C1V1 (activated by 546 nm) and ChrimsonR (activated by 665 nm) was 0.8:0.2. Hence the red emission from 808 nm has a minimal level of activation in ChrimsonR-expressing neurons (Supplementary Fig. 28). Moreover, using a mixture of monochromatic UCNPs with the same intensity of blue, green, and red emission bands as trichromatic UCNPs, we repeated the optogenetic experiment. Upon 980 nm excitation, a mixture of monochromatic UCNPs resulted in activation of all three light-preferring neurons; 808 nm excitation induced activation of both green-light-preferring and red-light-preferring neurons (Fig. 4h).

**Trichromatic modulation of mice locomotion driven by NIR optogenetics**. It is well-known that activities of premotor cortex (M2) neurons modulate locomotion in mice[34,35]. However, the contribution of excitatory and inhibitory neurons in M2 to the running speed is not clear. Using the trichromatic UCNPs, we are able to independently manipulate the activities of CaMKIIα and PV neurons and explore their role in locomotion. The protocol describing mouse behavior recording in virtual reality (VR) via UCNPs mediation can be found at Protocol Exchange[36]. We injected AAV-DIO-ChR2, AAV-CaMKIIα-C1V1 viruses, and UCNPs into M2 of PV-Cre mice (Fig. 5a–c). Two weeks after the injection, we first tested the neuronal responses to NIR light after penetrating the skull, skin, and brain tissue slices of different thicknesses. We demonstrated that 808 and 980 nm NIR light could activate neurons at 400 μm below the brain surface even with the skull and skin[37] (Fig. 5d). In the locomotion test, mice were head-fixed in front of a VR system[38] and trained to run in a

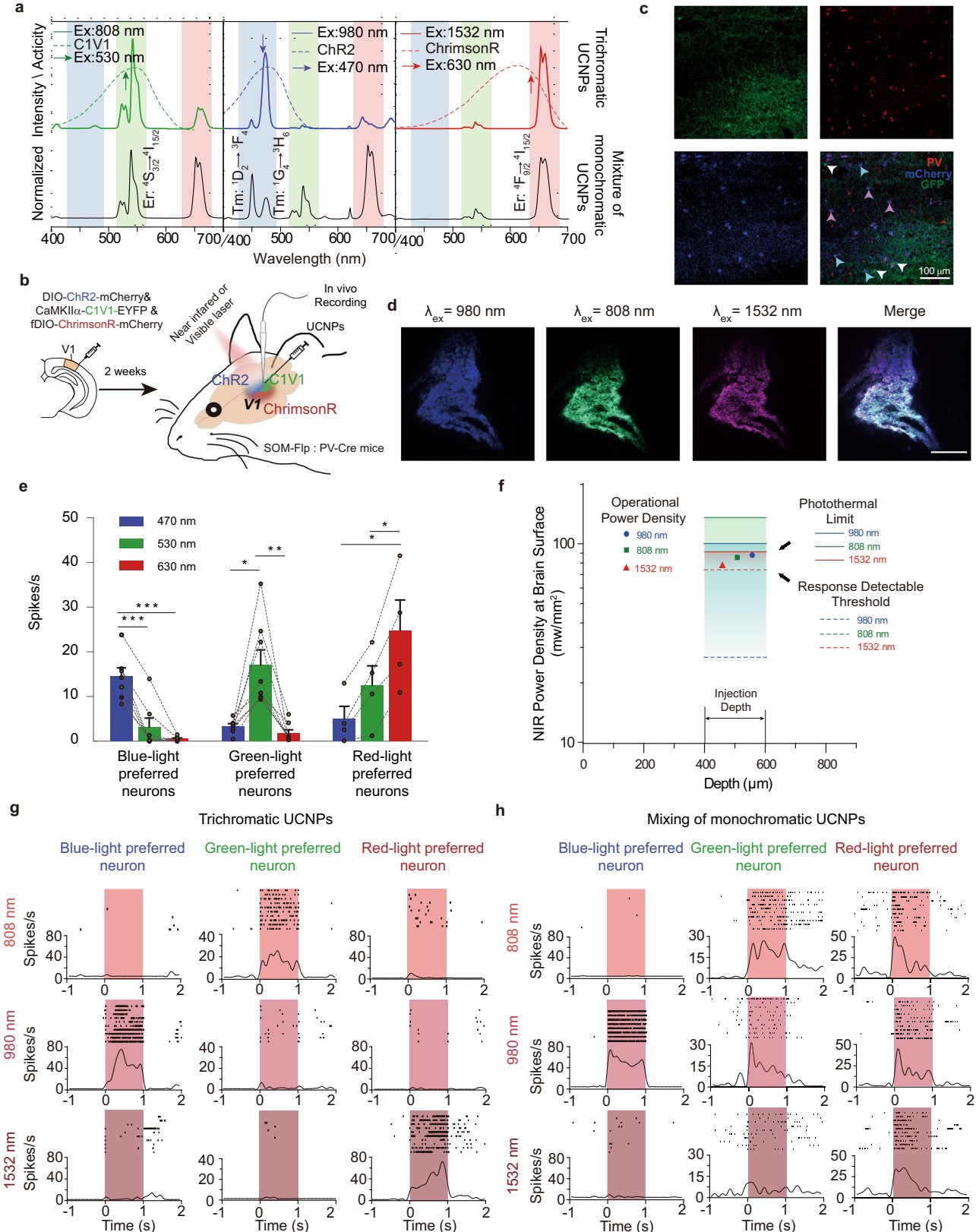

virtual one-dimensional corridor and two-dimensional open field environment with NIR light stimulation transcranially. Activation of CaMKIIα neurons in M2 by 808 nm laser resulted in an increase in the distance traveled, while activation of PV neurons by 980 nm laser resulted in a decrease in the distance traveled in both behavior tests (Fig. 5e, f and Supplementary Movie 2). When

both 808 and 980 nm lasers were illuminated onto the skull, distance traveled decreased, indicating that activation of PV neurons suppressed the movement of the mice despite CaMKIIα neurons were activated simultaneously. Activation of PV neurons here was proven to dominate locomotion behavior. The temperature increase was caused by the dual presentation of both

**Fig. 4 Trichromatic UCNPs-mediated in vivo optogenetic activation of three distinct neuronal populations. a** Emission spectra of trichromatic UCNPs (colored full line) and a mixture of three monochromatic UCNPs (black full line) under 808, 980, and 1532 nm NIR and activated spectra of three optogenetic proteins (dash line)[5, 28] (C1V1, ChR2, and ChrimsonR). Ex excitation wavelength. Power density is set at 20 mW/mm². Arrows indicate the wavelengths of the visible light lasers. **b** Schematic showing upconversion manipulated multicolor optogenetic experiment. The three viruses were injected into the primary visual cortex (V1) of somatostatin (SOM)-Flp:parvalbumin (PV)-Cre mice. UCNPs were injected into the same sites 2 weeks later and in vivo electrophysiology recordings were conducted in V1 thereafter. **c** Expression of C1V1-EYFP, ChrimsonR-mCherry, and ChR2-mCherry in CaMKIIα, PV, and SOM neurons respectively. PV neurons were labeled with PV-antibodies to distinguish ChR2-expressing PV neurons from ChrimsonR-expressing SOM neurons. White arrows indicate EYFP-expressing CaMKIIα neurons, light blue arrows indicate mCherry-expressing SOM neurons, and pink arrows indicate mCherry-expressing PV neurons. Scale bar, 100 μm. Immunohistochemistry was repeated four times with similar results observed. **d** Distribution of trichromatic UCNPs in V1. Fluorescent images were taken under 1532, 808, and 980 nm excitations. Scale bar, 200 μm. This experiment was repeated five times and similar results were observed. **e** Selectivity of opsins with 470, 530, and 630 nm laser excitation (0.44 mW/mm² for 470 nm, 0.70 mW/mm² for 530 nm, and 0.83 mW/mm² for 650 nm at the brain surface; 0.06 mW/mm² for 470 nm, 0.08 mW/mm² for 530 nm, and 0.14 mW/mm² for 650 nm at 400 μm depth, 1 s pulse width, 0.1 Hz, $n_{\text{blue-light preferred neurons}} = 7$ neurons in three mice, $n_{\text{green-light preferred neurons}} = 8$ neurons in two mice, $n_{\text{red-light preferred neurons}} = 4$ neurons in three mice, $^*P < 0.05$; $^{**}P < 0.01$; $^{***}P < 0.001$, one-way repeated-measures Analysis of variance (ANOVA) with Tukey's multiple comparison test). The exact $p$ values are listed in Supplementary Table 3 in the supplementary information. **f** Calculated allowable range of NIR power density at the brain surface. The operational power density is allowed for chromatic selectivity of optogenetic proteins and thermal limit as well as detecting the activation response. The operational NIR power densities are used in the electrophysiology experiment. **g** Representative optogenetic responses in blue-light preferred (left), green-light preferred (middle), and red-light preferred neurons stimulated by NIR light with trichromatic UCNPs at 980, 808, and 1532 nm, respectively (89.1 mW/mm² for 980 nm, 86.4 mW/mm² for 808 nm, and 76.8 mW/mm² for 1532 nm at brain surface; 22.8 mW/mm² for 980 nm, 15.3 mW/mm² for 808 nm, and 20.2 mW/mm² at 400 μm depth, 1 s pulse width, 0.05 Hz). The low spontaneous firing rate was due to a relatively deep anesthesia state). **h** Same as in (**g**) but for a mixture of three monochromatic UCNPs. Source data are provided as a Source Data file.

---

808 nm (light intensity was 59.1 mW/mm² at the brain surface) and 980 nm (light intensity was 38.9 mW/mm² at the brain surface) NIR light at the skull surface was 4.7 °C (Supplementary Fig. 25c). We also provided proof-of-principle evidence for three-color optogenetic manipulation of motor behavior using the mice in Fig. 4b (Supplementary Fig. 29). The average running distance increased under 808 nm (activation of CaMKII neurons) or 1532 nm (activation of SOM neurons) laser but decreased under 980 nm (activation of PV neurons) laser.

Such tunable colors have the potential for a gradient manipulation of the neural activity and hence the behavioral output. The activity of excitatory CaMKIIα neurons was proved to be regulated by PV inhibitory neurons[35]. We activated C1V1-expressing CaMKIIα neurons by 808 nm laser and simultaneously changed the power density of the 980 nm laser, which modulated activities of ChR2-expressing PV neurons, to manipulate the locomotion behavior. Neuronal responses were measured under series of excitation power densities. The firing rates of CaMKIIα neurons decreased as 980 nm laser intensity increased (Fig. 5g). Distance traveled was increased by 808 nm laser's activation and decreased gradually when 980 nm laser intensity increased both behavior tests (Fig. 5h–j), indicating a gradient modulation of mice motor function by dynamic NIR upconversion tuning.

Our UCNP enabled trichromatic-optogenetic dissection of neural circuits. We next explored how co-activation of CaMKIIα, PV, and SOM neurons in M2 affects motor behavior. The challenge for trichromatic activation is the temperature increase when 808, 980, and 1532 nm are turned on simultaneously. A recent study showed that temperature increase for over 1–2 °C is sufficient to affect specific neuron cells like medium spiny neurons and to induce change in behavior[39]. We explored the stimulation condition that minimized the temperature increase and found that the temperature rise was below 1 °C when the power of NIR lasers were 0.8 mW/mm² for 808 nm, 1.27 mW/mm² for 980 nm, and 1.27 mW/mm² for 1532 nm lasers at 5 Hz with 0.1 s pulse duration for 120 s total duration (Supplementary Fig. 30a). Using these trichromatic stimulation conditions, 808 nm laser or 1532 nm laser resulted in increased running distances while stimulation of 980 nm laser resulted in decreased running distance, which was similar to previous results in locomotion test on one-dimensional corridor scene (Supplementary Fig. 31).

When 808, 980, and 1532 nm lasers were all illuminated onto the skull, the average distance traveled was smaller than control, indicating that activation of ChR2-expressing PV neurons dominated the modulation in this test. The reason for this dominating effect may lie in the difference of NIR power density and conversion efficiency as well as the activation thresholds of ChR2, C1V1, and ChrimsonR, in which PV neurons expressing ChR2 is the most of which being activated. The depolarization time $\tau_{\text{off}}$ of optogenetic protein may also matter. Besides, there is a chance that PV neurons suppress activities globally while SOM and CaMKIIα neurons function locally, which may also count for the observed domination.

Very few optogenetic studies require prolonged stimulation for over 100 s. So we further explored the feasible conditions meeting the requirements of temperature rise (below 2 °C) with higher power when the pulse frequency is 10 or 20 Hz for a shorter total duration of 10 or 20 s. By decreasing the pulse duration to 25–50 ms and increasing the stimulation frequency to 10–20 Hz, the upper boundary for light intensity is increased by 87.5% at 808 nm and 116.5% at 980 nm, and 25.2% at 1532 nm. (Supplementary Fig. 30b, c).

## Discussion

We have overcome a few technical hurdles in developing trichromatic UCNPs with a related optogenetic application, providing a proof-of-principle demonstration for multicolor upconversion optogenetics. First of all, the spectral separation in our trichromatic UCNPs prevented the cross talk in multicolor optogenetic excitation that single or mixture of conventional UCNPs could not independently and precisely excite opsins due to their multiple emission peaks. Secondly, we resolved the issue of spectral overlap between ChrimsonR and ChR2/C1V1 by finding the combination of light intensities that enabled the independent activation of different optogenetic proteins. We have also carefully measured the safe light intensity for NIR stimulation to prevent overheating in the brain tissue. It is important to note that the current version of UCNPs is not yet suitable for the activation of deeper brain regions. Further improvement in the efficiency of UCNPs is necessary to push the limit of activation depth.

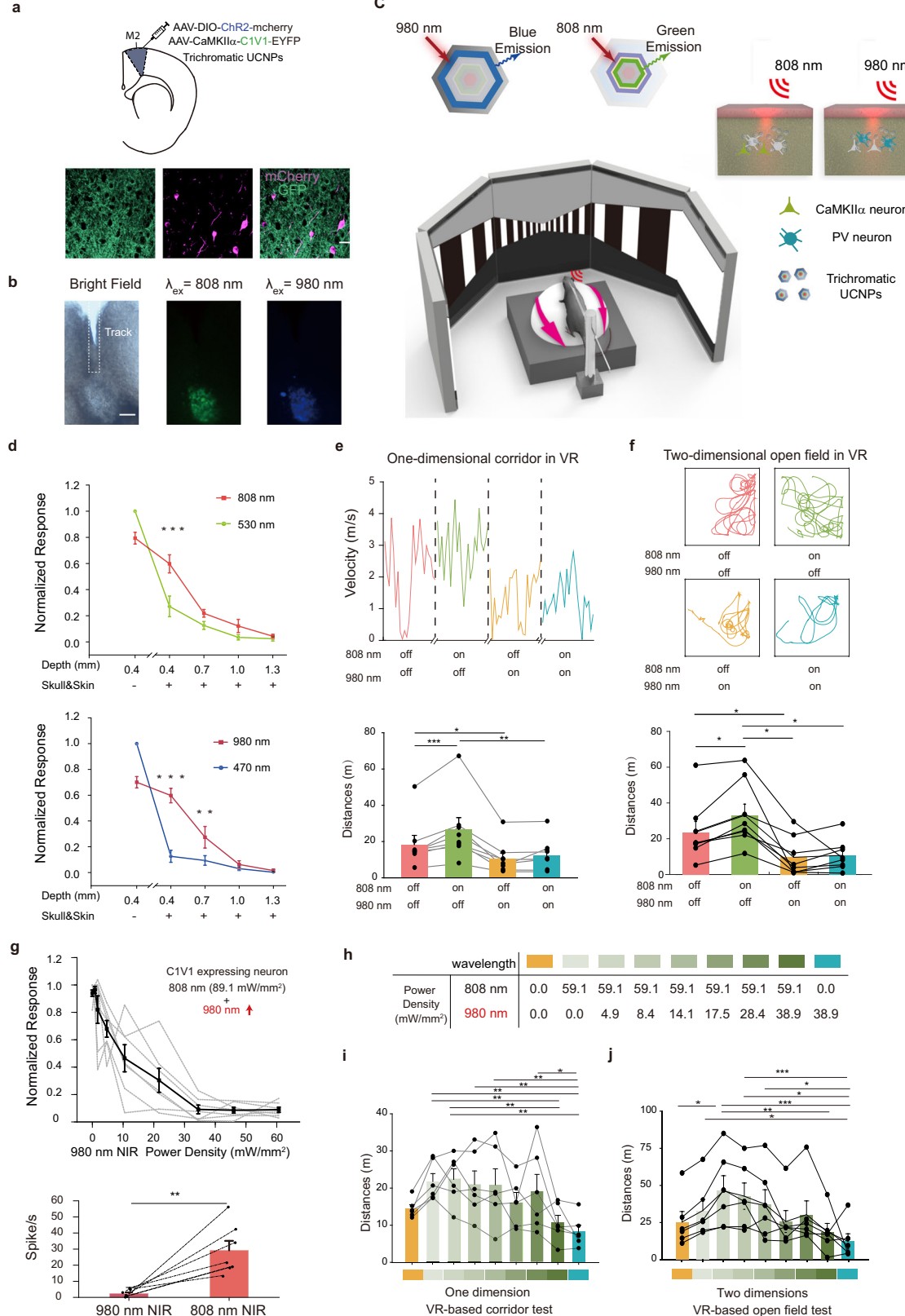

For the design of the UCNPs, previous reports have demonstrated two-color orthogonal blue/green upconversion with sensitizer $Yb^{3+}/Nd^{3+}$. In those previous nanostructures, both core and shell have the same sensitizer $Yb^{3+}$ which can be excited by 980 nm light. The responsive dichromatic emission was realized by a higher doping concentration of $Yb^{3+}$ in the outer layers for absorbing 980 nm photons. In our structure, different types of sensitizers are used, namely $Er^{3+}/Yb^{3+}/Nd^{3+}$. To construct responsive trichromatic emission, we investigated the dissipation process between sensitizer of erbium and ytterbium as well as erbium and neodymium and found that energy extraction induced dissipation in the outer shell is a critical factor for

**Fig. 5 Transcranial optogenetic gradient multicolor manipulation of mouse locomotion behavior using UCNPs. a** Schematic showing the stereotaxic injection of AAV-DIO-ChR2-mCherry, AAV-CaMKIIα-C1V1-EYFP viruses, and UCNPs in M2 (top) and corresponding fluorescence image showing the expression of ChR2-mCherry and C1V1-EYFP in PV and CaMKIIα neurons, respectively in M2. Scale bar, 50 μm. **b** Distribution of trichromatic UCNPs in M2. Fluorescent images were taken under 808 or 980 nm excitation. Scale bar, 200 μm. Immunohistochemistry was repeated twice and similar results were observed. **c** Schematic illustrations of head-fixed PV-Cre mouse running in VR System. NIR laser (four conditions: none, 808, 980 nm, or both) was transcranially illuminated onto M2 for 120 s (100 ms pulse width, 5 Hz). **d** Neuronal responses of visible laser (530, 470 nm, 1 s pulse width, 0.1 Hz) and NIR laser (808, 980 nm, 1 s pulse width, 0.05 Hz) with skull, skin, and different thicknesses of brain tissue on the top ($n_{808nm}$ = 4 neurons in two mice, $n_{980nm}$ = 6 neurons in two mice, **$P < 0.01$, ***$P < 0.001$, one-way repeated-measures ANOVA with Tukey's multiple comparison test). **e** Running speed of one mouse in a one-dimensional corridor in VR system under 808 and 980 nm NIR excitations and the corresponding normalized average running distances ($n$ = 8 mice, three trials for each mouse, *$P < 0.05$, **$P < 0.01$, ***$P < 0.001$; two-way repeated-measures ANOVA with Tukey's multiple comparison test). **f** Traces of locomotion movement in a two-dimensional open field in VR system under 808 and 980 nm NIR excitations and the corresponding normalized average running distances ($n$ = 8 mice, three trials for each mouse, *$P < 0.05$; two-way repeated-measures ANOVA with Tukey's multiple comparison test). **g** (Top) Normalized CaMKIIα neurons' responses ($n$ = 7 neurons in four mice) evoked by 808 nm laser (power density = 86.4 mW/mm$^2$ at brain surface, 1 s pulse width, 0.05 Hz) together with an increasing power density of 980 nm laser (power density from 0 to 38.9 mW/mm$^2$ at brain surface, 1 s pulse width, 0.05 Hz), gray lines represent responses of each C1V1-expressing neuron, respectively. (Bottom) Responses of recorded neurons under 980 and 808 nm NIR (**$P = 0.005$, two side paired $t$-test). **h** Power densities for different conditions were shown in terms of color. **i** Normalized running distances of mice ($n$ = 6 mice, three trials for each mouse) in a one-dimensional corridor in a VR system. (*$P < 0.05$, **$P < 0.01$; two-way repeated-measures ANOVA with Tukey's multiple comparison test). **j** Normalized running distances of mice ($n$ = 6 mice, three trials for each mouse) in a two-dimensional open field in a VR system. (*$P < 0.05$, **$P < 0.01$, ***$P < 0.001$; two-way repeated-measures ANOVA with Tukey's multiple comparison test). The exact $p$ values are listed in Supplementary Table 3. Data were presented as mean ± s.e.m. Source data are provided as a Source Data file.

blocking specific excitation wavelength. Based on the strategy of enhancing dissipation, incident excitation photon fluxes can be precisely regulated to specific activators in a single nanoparticle to realize the NIR excitation-specific non-cross talk trichromatic upconversion emissions. We believe such an energy dissipation strategy for the mediation of excitation energy distribution in the core/shell structure would benefit the nano-engineering of core/shell lanthanide-doped nanoparticles.

Developing next-generation optogenetic tools with UCNPs will facilitate the dissection of complicated brain circuits as well as the application in clinical conditions. By elaborate design and detailed testing, we managed to independently modulate multiple neuronal types with NIR lights and UCNPs-mediated multicolor optogenetics. One long-term goal of neuroscience is the recording activities of distinct neuronal populations in the highly complicated and holistic brain network, as well as the manipulation of selective neurons to understand what leads to complicated behaviors including vision, emotion, learning, and memory. Such a task is challenging, given that the anatomical and functional structure of the brain network is often hierarchical and reciprocal, with multiple inputs from different brain regions and multiple outputs to others. When the brain is at work, information was processed within the brain network multidirectionally. Therefore, studying the brain circuits with one optogenetic protein would only allow unidirectional dissection of the behavior. Developing next-generation multicolor optogenetic tools with UCNPs will facilitate the multidirectional dissection of complicated brain circuits. Specifically, how inputs with varying levels of coherence from multiple brain regions lead to different emotional outcomes, as well as how activities of ensembles at different hierarchical levels act collectively to contribute to the integration of sensory information.

A recent annual report of deep brain stimulation (DBS) think tank has mentioned advances in optogenetics, which have now emerged as a tool for comprehending neurobiology of diseases and inspired cutting-edge DBS[40]. Transcranial optogenetic photoactivation opens up the possibility to explore therapeutic interventions for neurological diseases. With the development of multicolor transcranial optogenetics, we can now target multiple brain areas simultaneously. For instance, the bed nucleus of the stria terminalis (BNST) circuit including the hypothalamus, parabrachial nucleus, and the ventral tegmental area is thought to be involved in anxiety[41]. Coherent manipulation of the BNST circuit will potentially enable the development of precise treatments.

There is also some limitation in our study. The first is that laser power needs to be kept at a restricted range due to overheating issues combined with the response threshold, which naturally occurs for all the other NIR-related studies. Though we provided stimulation conditions that meet the requirements in this work, the temperature rise still needs to be measured in advance when transferring this technique to other studies. The second is about protein chromatic selectivity, we address the spectral overlap between ChrimsonR and ChR2/C1V1 by trimming light intensities to give an independent activation. Advance in opsins with less spectral overlap will be an effective improvement to this technique. For example, the newly reported HfACR1 is red-shifted farther than ChrimsonR[42]. We believe that red-shifted channelrhodopsins with better spectral segregation would further improve the multi-chromatic optogenetic neural circuit dissection in the near future.

In summary, our study reveals the regulation of photon energy flux and shows the feasibility of UCNPs based multicolor optogenetics, which enables independent investigation of multiple neuronal populations in awake-behaving animals with deep penetration. Further works can be promoted to wearable, implantable devices, or auto-sighting of lasers in order to free and deep control. We envision it will provide a first-order method for the characterization of hierarchical signal processing streams and exhibit potential application in the tuning of animal behaviors as well.

## Methods

**Synthesis of nanoparticles**. The fabrication of core-shell nanoparticles was carried out by the thermal decomposition method followed by epitaxial growth via a hot-injection protocol. Additional experimental details are provided in the Supplementary Information.

**Surface modification of nanoparticles**. The synthesized nanoparticles were transferred into the liquid phase with the method previously reported by Yao et al[43]. Typically, 1 mL of oleic acid capped UCNPs in cyclohexane (10 mg mL$^{-1}$) was re-dispersed into 1 mL chloroform and mixed with 1 mL chloroform solution containing 12.5 mg DSPE-PEG-OCH3 in a 250 mL flask. Then the solvent were rotated by 100 rpm and evaporated for 20 min. The resulting mixed film was heated at 80 °C for 5 min to completely remove the solvent. Then the film was hydrated by 20 mL

water with vigorous sonication. The solution was transferred to a microtube and excess lipids were purified by ultracentrifugation (15,000 rpm, 10 min). The subsequent sediment was re-dispersed into 1 mL saline solution.

**Characterization of nanoparticles.** Transmission electron microscopy (TEM), high-resolution transmission electron microscopy (HRTEM), and high-angle annular dark field imaging in the scanning TEM (HAADF-STEM) observations were performed on JEM-2100F transmission electron microscope with an accelerating voltage of 200 kV equipped with a post-column Gatan imaging filter (GIF-Tri-dium). The photoluminescence emission spectra were recorded on HORIBA Fluorolog equipped with the external 980 and 808 and 1532 nm semiconductor laser excitation sources (Changchun New Industries Optoelectronics Tech. Co.Ltd.). NIR dual-excitation was carried out through a converged light path with dichroscopes. Unless otherwise specified, all spectra were collected under identical experimental conditions. The spectra data were edited by OriginPro8.

**Single-particle imaging and in situ spectral analysis.** Dilute UCNPs were dispersed on the coverslip and imaged by spectrograph-coupled optical microscopy under the excitation of tightly focused CW lasers of 1532, 980, and 808 nm, respectively. Images were obtained by using an Olympus IX71 microscope equipped with the SP 2360 spectrometers and Pro EM CCD camera (Princeton Instruments Inc.). The 980, 808, and 1532 nm CW lasers were used as the excitation sources combined with a 700 nm short-pass optical filter (Chroma Corp). Luminescence spectra of a single particle at the same position of the image was obtained from a CCD spectrophotometer with wavelength scanning. Upconverting luminescent images of the samples showed homogeneously and randomly distributed spots within diffraction limit on the substrate. From the emission spectra of over 100 individual luminescent spots, monochromatic channel intensities of red, green, and blue under corresponding NIR excitations were calculated. Channel intensity was derived from the calculation of average intensity at the specific wavelength ranges (438–480 nm for blue, 507–560 nm for green, and 640–700 nm for red). The scattering plots of monochromatic channel intensities under corresponding NIR excitation showed well maintained trichromatic emissive property at a single-particle scale.

**Synchronous galvanometric scanning.** Three CW diode lasers (Changchun New Industries, China) (980, 808, and 1532 nm), which connected to a computer with an RS232 cable, were aligned and directed into a fast scanning 3D galvanometer. The power of each laser was adjusted through changing the input current by RS232 and supporting software. The scanning of the laser beams was controlled simultaneously by Cyberlease scanning software.

**NIR upconversion efficiency measurement.** Trichromatic UCNPs (0.75 mg) were dropped on a glass slide. NIR lasers tilts to excite the UCNPs with fixed fiber. The power meter (PM100D, S310C; Thorlabs) was set perpendicularly to the excitation light plane to measure the emission intensity filtered by a 700 nm short-pass filter. Conversion yield was calculated by dividing visible light emission output power with corresponding input power.

**Animals.** Six-week-old C57BL/6 J male mice were purchased from Shanghai Laboratory Animal Center, Chinese Academy of Sciences. Male PV-Cre mice (Jackson lab, US. Stock #008069) and SOM-Flp mice (Jackson lab, US. Stock #031629) aged from 6-week to 14-week were used in experiments. Mice were housed on a 12-h light–dark cycle with food and water ad libitum at 23 °C, and the room humidity was controlled around 40%. All protocols for animal experiments were in strict accordance with the National Institutes of Health Guide for the Care and Use of Laboratory Animals and approved by the Animal Care and Use Committee of Fudan University.

**Immunohistochemistry.** Mice were perfused with PBS and then 4% PFA (Sigma, US) after recording. Brains were taken out and fixed in 4% PFA overnight at 4 °C, then dehydrated by 30% sucrose in PBS (324 mM $Na_2HPO_4 \cdot 12H_2O$, 76 mM $NaH_2PO_4 \cdot 2H_2O$) for 48 h. Brains were embedded in optimal cutting temperature (OCT) compound and stored at −80 °C. Slices were cut into 30 μm sections by a cryostat (LEICA CM1950; Germany) and washed with Tris-buffered saline for five times (15 min each time) to remove OCT. Slices were permeabilized in 0.5% Triton-X-100 for 30 min and blocked by 10% donkey serum for 2 h at room temperature, followed by primary antibody (GFP, Cat# GFP-1020, 1:800, Avēs Lab Inc, US; mCherry, Cat# AB0081-200, 1:2000, SICGEN, Portugal; PV, Cat# PV 25, 1:800, Swant Inc, Swiss; SOM, Cat# ab64053, 1:200, Abcam, UK; CaMKIIα, Cat # ab22609, 1:200, Abcam, UK; Iba1, Cat# ab5076, 1:500, Abcam, UK; GFAP, Cat# G6171, 1:1000, Sigma, US) hybridization overnight at 4 °C. The slices were washed for another five times (5 min for each time) in Tris-buffered saline, and then secondary antibody (donkey anti-chicken 488, Cat# 705-545-003, 1:300; donkey anti-goat 594, Cat# 705-585-003, 1:200; donkey anti-rabbit 594, Cat# 711-585-152,1:200; donkey anti-rabbit 647, Cat# 711-605-152, 1:200; Jackson ImmunoResearch, US) was added and the slices were incubated at room temperature for 2 h in the dark. Fluorescently stained samples were then washed three times (15 min

for each time), then stained in 4′,6-diamidino-2-phenylindole (DAPI) (Sigma, 200 ng ml⁻¹) solution for 7 min followed by three washes (15 min for each time) and mounted with AQUA-MOUNT (Thermo Scientific, US).

**Viruses and UCNPs injection.** AAV2/9-EF1a-double floxed-hChR2(H134R)-mCherry-WPRE-HGHpA, AAV2/9-hSyn-hChR2(H134R)-mCherry, AAV2/9-mCaMKIIa-C1V1(t/t)-TS-EYFP, AAV2/9-Ef1a-DIO-C1V1-EYFP, AAV2/9-Syn-Cre, AAV2/9-hSyn-ChrimsonR-tdTomato-WPRE-SV40Pa, and AAV2/9-hEF1a-fDIO-ChrimsonR-mCherry were purchased from Taitool (Taitool, China). Mice were anesthetized by 2% chloral hydrate (20 mL per 100 g body weight), and supplemental doses were subcutaneously delivered as required. Mice were placed in a stereotaxic frame (RWD, China), and the head was fixed on a setup by nose clip and ear bars. One centimeter long incision was made in the middle of the scalp, the exposed skull overlying V1 (1.5–3.5 mm M/L, 0.5–2.5 mm A/P, 0–0.6 mm D/V to lambda suture) or M2 (0.5 mm M/L, 1 mm A/P, 0.5 mm D/V to lambda suture) was thinned by the cranial drill in three locations. Glass pipette filled with viruses was inserted to injection sites around 400–500 μm. A volume of 69 nL of viruses was injected at a rate of 13.8 nL/min at each location by nano ject II (Drummond Scientific Company, US), and the pipette was kept in the brain for 15 min after injection at each location. The scalp was then sutured. UCNPs injections were similar to virus injections, except that the glass pipette was filled with 400 nL UCNPs saline solution, and the injection rate was 69 nL/min.

**Whole-cell patch-clamp electrophysiology in brain slice.** For slice recording, slices were perfused with artificial cerebrospinal fluid (ACSF) (119 mM sodium chloride, 2.5 mM potassium chloride, 26 mM sodium bicarbonate, 12.5 mM D-glucose, 5 mM HEPES, 2 mM calcium chloride, 2 mM magnesium chloride, and 1.25 mM sodium phosphate monobasic monohydrate (pH 7.3)) and bubbled with carbogen (95% oxygen and 5% carbon dioxide). Thick-walled borosilicate glass (Sutter BF150-86-10) electrodes were pulled (Sutter P97) to a resistance of 5–7 MΩ. The electrodes were filled with internal solution (105 mM potassium gluconate, 5 mM potassium chloride, 0.5 mM calcium chloride, 2 mM magnesium chloride, 5 mM ethylene glycol-bis (2-aminoethylether)-N,N,N′,N′-tetraacetic acid, 2 mM adenosine 5′-triphosphate magnesium salt, 0.5 mM guanosine 5′-triphosphate sodium salt hydrate,7 mM phosphocreatine disodium salt hydrate, and10.0 mM HEPES. pH 7.2, osmolarity 280 mOsm). Visible light was filtered to 470, 546, and 665 nm through a microscope. Light power was controlled through X-cite Illuminators and light pulse was adjusted to 1 s. NIR lasers (808, 980, and 1532 nm) were delivered through an optical fiber. The UCNPs diffused in the which solution couldn't maintain a stable concentration in the recording chamber as those in in vivo experiments. UCNPs illuminant were thus fixed in the form of transparent polymethylpropanamide (PMMA) film. Specifically, for preparation, 1 mL UCNPs stock solution were re-dispersed to 4 mL chloroform with 0.40 g dissolved PMMA. The solid UCNP film was then obtained by dripping the solution on the cover glass and completely evaporating the chloroform solvent. The brain slice was then placed onto the UCNP film and the estimated distance between the film and recorded neurons in the brain slice was 0.25 mm. Whole-cell patch-clamp recordings were made using Multiclamp 700B amplifier and a Digidata 1440 digitizer. Data were analyzed using Clampfit (Molecular Devices). Data were normalized by the maximal response in each cell.

**Temperature measurement on the brain surface.** A mouse was anesthetized and head-fixed in a stereotaxic frame (RWD, China) by nose clip and ear bars. A one-centimeter-long incision was made in the middle of the scalp to expose the skull or cortex. Optical fiber was placed 2 mm away from the surface, NIR light at 808, 980, and 1532 nm were delivered with pulse durations (1 s stimulus–19 s interval in in vivo recording and 0.1 s stimulus–0.1 s interval in behavioral experiments) controlled by custom-designed scripts in Arduino. The heat effects of the laser sources were recorded on a professional infrared thermal imaging camera (FLIR Thermal CAM A300). Further analysis was conducted with custom codes in Matlab.

**Transmission electron microscope of brain tissue.** Three weeks after UCNPs injection, the mouse was induced and the brain was dissected after perfusion of saline solution. Then the brain was transferred to 2.5% Glutaraldehyde as quickly as possible and stored at 4 °C overnight. The fixed tissue was cut into a tiny cube and processed for further observation. Several 70 nm ultrathin sections were prepared for electron microscope imaging.

**Calculation of converted visible light intensity.** To confirm that the UCNPs-mediated IR-to-visible light emission was within the optogenetic selectivity criteria, we calculated the light intensities of the converted visible light (Supplementary Fig. 14). NIR light was irradiated onto the surface of V1, the light intensities of 808, 980, and 1532 nm were set as 86.40, 89.10, and 76.80 mW/mm² at the brain surface, respectively. Corresponding converted visible light intensities were 0.40 mW/mm² for blue light, 0.13 mW/mm² for green, and 0.02 mW/mm² for red at the recording site around 400 μm below the brain surface and were within the optogenetic selectivity criteria.

**Simulation of upconversion emission upon transcranial NIR irradiation**. Transmission of NIR light through the brain slices was modeled using the Kubelka–Munk theory. For simplify the model, it was assumed that reflection is constant and absorption is negligible over the thickness of the sample. This model fits well in previous visible light and monochromatic NIR light in brain tissue transmittance study[44]. For simple scattering media,

$$T = \frac{1}{S \cdot z + 1}$$

where T is transmission fraction, S is the scatter coefficient, and z is the thickness of the sample. The focal distance $\rho$ of fiber outlet can be calculated as

$$\rho = r \sqrt{\left(\frac{n}{NA}\right)^2 - 1}$$

where $n$ is the index of refraction of tissue, $r$ and NA is the radius and numerical aperture of the optical fiber, respectively. The geometric component of emitted light's decrease at a given distance from the fiber can be calculated,

$$\frac{I(z)}{I(0)} = \frac{\rho^2}{(z + \rho)^2}$$

Taking into account both scattering and geometric losses, the complete expression of penetrated light intensity is

$$I(z) = \frac{\rho^2}{(Sz + 1)(z + \rho)^2} I(0)$$

The scattering coefficient of different NIR wavelengths can be derived from a previously reported experimental formula[5],

$$S = 4.72\lambda^{-2.07}$$

where $\lambda$ is the wavelength in microns and S stands for the scatter coefficient of NIR lights in brain tissue within the 700–1700 nm range. In our study, the radius and numerical aperture of fiber are 400 μm and 0.22. The power density of each NIR laser-measured without tissue sample is 153.64 mW/mm² of 808 nm and 125.57 mW/mm² of 980 nm. Index of brain tissue is tested to be ~1.36. The intensity I(z) can then be predicted with different tissue depths z.

**Neuronal recording with tissues**. To compare the penetration efficiency of NIR light with that of visible light, we first found neurons responding to both 470 nm laser/980 nm NIR laser light or to 530 nm laser/808 nm NIR laser light. We used a custom-design pole holding the optical fiber 2 mm away from the recording site. Baseline responses were first recorded at 400–500 μm. We then placed a 1 mm × 1 mm skull with skin attached between the recording site and the optic fiber. About 300-μm-thick brain slices (from mice perfused without PFA) were added on the skull with skin attached. Neuronal responses were recorded in all conditions.

**In vivo electrophysiology recording**. Mice were first head-fixed on stereotaxic setup (Thorlabs Inc, US), and a respiratory mask was placed around the nose and mouth to deliver 1.5% isoflurane for anaesthetization during surgery and turned down to 0.5% isoflurane during electrophysiology recording. The eyes were covered with erythromycin ointment to block the environmental light. After removing the scalp, a craniotomy window was created stereotaxically on V1. The dura was carefully removed. The silicon electrode (A4x8-5mm, Neuronexus, US) was slowly inserted into V1 at a depth of about 0.4–0.6 mm by a micromanipulator (Scientifica, US). Electrical signals were recorded at 30 kHz and amplified ×200 by a multichannel data acquisition system (Bio-Signal Technologies, China). Spikes were high-pass filter at 300 Hz, detected as events exceeding a threshold of 5× s.d. below the noise, raster plot and peri stimulus time histogram (PSTH) was made in spike2 (Cambridge Electronic Design, UK) and MATLAB (Mathworks, US). The spike waveforms were sorted off-line by OfflineSorter (Plexon, US). The craniotomy was filled with fresh warm sterile buffered saline (150 mM NaCl, 2.5 mM KCl, 10 mM HEPES, pH 7.4) throughout the entire recordings.

**Mouse behavior in VR system[38]**. Titanium alloy head bar (about 5 mm × 30 mm) was implanted opposite to the virus injection side with two skull screws and dental cement (C&B Metabond, Parkell, US) before behavior training. After 3-day recovery, mice were trained to run on a jetball with a one-dimension corridor or two-dimension open field scene displayed on VR system (JetBall TFT, Phenosys Inc) screens. The one-dimension corridor was 200 m long and the two-dimension open field was 20 m × 20 m. Head-fixed mice ran on the jetball for 2 min per session under the conditions of no NIR light, NIR light at 808, 980, 808&980 nm, and 808&980&1532 nm, respectively (pulse duration = 0.1 s, interval = 0.1 s, 5 Hz, power densities of NIR laser were 59.10 mW/mm² for 808 nm, 38.90 mW/mm² for 980 nm, and 0.8 mW/mm² 808 nm, 1.27 mW/mm² 980 nm, 1.27 mW/mm² 1532 nm NIR laser for trichromatic modulation). For independent multicolor modulation, the power density of NIR light at 980 nm was increased (power density from 0 to 38.9 mW/mm² at brain surface) while the power density of NIR light at 808 nm was 59.10 mW/mm². Mice would rest for more than 5 min for the next trials, and the trail orders were randomly chosen. Running distances of mice were recorded and measured.

**Statistical analysis**. Statistical analyses were performed in Sigmaplot (SYSTAT, USA). Data distributions were tested for normality using the Kolmogorov–Smirnov normality test. For comparison of multiple groups, data were analyzed using one-way repeated-measures ANOVA followed by Tukey post hoc analysis. Two-way repeated-measures ANOVA followed by Tukey post hoc was used for behavioral data analysis in Fig. 4e, f. $P$ value <0.05 was considered statistically significant. Data were presented as the mean ± s.e.m. A summary of the statistical analysis in the manuscript was shown in Supplementary Table 2.

**Reporting Summary**. Further information on research design is available in the Nature Research Reporting Summary linked to this article.

## Data availability
The luminescence, parameter condition test, and optogenetic stimulation data generated in this study are available in the figshare database with identifiers https://doi.org/10.6084/m9.figshare.16529412 and https://doi.org/10.6084/m9.figshare.16529508. Source data underlying figures are provided with this paper. Source data are provided with this paper.

## Code availability
The custom code for laser control is available at https://github.com/Izulu/trichromatic-optogenetic.

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

## Acknowledgements

This work was supported by the National Key R&D Program of China (2017YFA0207303), National Science Foundation of China (21725502, 31771195, 81790640, 21701027, 21875043, and 82021002), Key Basic Research Program of Science and Technology Commission of Shanghai Municipality (20490710600, 20YF1402200, 20JC1411700, 19490713100, 20S31903700), Key scientific-technological innovation research project by Ministry of Education, Natural Science Foundation of Shanghai (18ZR1404600), Shanghai Municipal Science and Technology Major Project (2018SHZDZX01), ZJLab and Shanghai Sailing Program (17YF1401000).

## Author contributions

F.Z., X.M.L. and X.L. formed the original concept. X.L., H.C. and Y.W. contributed equally to this work. X.L. and Z.Z. synthesized the nanoparticles. X.L., H.Z. and X.M.L. developed the optical system. H.C., Y.W., Y.S., B.Y. and X.L. designed and conducted animal experiments. X.L., H.C. and Y.W. were primarily responsible for data collection. X.L., H.C., X.M.L., D.Z., S.W., J.Z. and F.Z. analyzed the results, prepared the manuscript, figures, and supplementary information. All authors contributed to the discussion and editing of the manuscript.

## Competing interests

The authors declare no competing interests.
