## [Peer Review File · Nature Communications]

Reviewers' Comments:

Reviewer #1:

Remarks to the Author:

Optogenetic technology has become a powerful toolset for manipulating brain functions, by targeting on genetically modified neurons expressing light sensitive and wavelength specific opsins. Although blue light sensitive ChR2 is widely used, it is highly demanded to develop a palette of opsins with different excitation spectra, so that different groups of neurons can be separately labeled (*Nature. Methods* **11**, 338-346 (2014)). Meanwhile, advanced optical technologies are needed to deliver light at corresponding wavelengths, to selectively interact with these opsins and excite or inhibit these neurons and precisely control neural activities. In this paper, a new type of upconversion nanoparticles (UCNPs) is developed to realize trichromatic emission with infrared illumination. By carefully controlled chemical synthesis, the UCNPs are constructed with multiple coatings, and the energy transfer mechanism determines that the emission colors (blue, green or red) can be tuned dependent on the irradiation wavelengths (980 nm, 808 nm and 1532 nm). These UCNPs are injected into the rodent brains with co-expressed ChR2, C1V1 and ChrimsonR opsins targeting on different neuron types. The authors demonstrate that the manipulations of multiple neuronal populations can be achieved by varying the infrared irradiation wavelengths. In terms of materials synthesis and the unique optical properties of the UCNPs, the results are beautiful and very intriguing. However, my major concerns are mostly in the biological experiments. Detailed comments are listed below:

The spectral responses of ChR2, C1V1 and ChrimsonR are plotted below, with data extracted from:

Klapoetke, N. C. *et al.* Independent optical excitation of distinct neural populations. *Nat. Methods* **11**, 338-346 (2014).

Although the paper demonstrate three different emission colors (475 nm, 540 nm and 650 nm) can be excited by different infrared sources (980 nm, 808 nm and 1532 nm) using the designed structure of UCNPs. However, the inherent spectral overlap of ChR2, C1V1 and ChrimsonR cannot be bypassed. It is clear that 475 nm emission can excite both ChR2 and C1V1 (response ratio ~ 1:0.7), and 540 nm emission can excite both C1V1 and ChrimsonR (response ratio ~ 1:0.7), very effectively. It is difficult to believe that these three opsins can be distinctively resolved using the tri-color emission.

To validate the concept for multi-color modulation, the neuronal activities should be recorded at the cellular level with a patch clamp, using cultured neurons or brain slices. For example, co-culture C1V1-expressing cells with the UCNPs, then demonstrate that the cells can only be excited by 808 nm, but not by 980 nm or 1532 nm. The same for ChR2- and ChrimsonR-expressing cells.

Fig. S5 shows that the UCNPs have stable PL output for up to 120 mins in solutions. This is not enough. The UCNPs are injected into the animal brain for more than a week (Fig. S13). How to validate that these UCNPs are stable in the brain for such a long time?

Electron micrographs should be provided, to see the distributions of these UCNPs within the brain tissue. Do they spread outside the cells, or are they uptaken by the cells?

Fig. 4c has very low quality. It is highly recommended that the authors prepare a better brain slice to show more neurons with fluorescence labels, in order to show that three different opsins (ChR2, C1V1 and ChrimsonR) are really expressing in three distinct neuron types, with minimal co-expressions.

Fig. 4d shows the distributions of UCNPs in the brain tissue. Unlike viruses that can target on specific cell types or regions, these UCNPs do not have any cell specificity. The natural diffusion of UCNPs should result in an isotropic distribution. Why do these images present an irregular shape of fluorescence? What limits the diffusion of these UCNPs?

Line 205 in supplement, “Corresponding converted visible light intensities were 0.40 mW/mm² for blue light, 0.13 mW/mm² for green and 0.02 mW/mm² for red at the recording site around 400 μm below the brain surface and were within the optogenetic selectivity criteria.”. To the reviewer’s knowledge, the typical excitation power density required for optogenetic stimulation is around 1-10 mW/mm². It seems that the power generated by these UCNPs is too low for effective modulation.

In supplement, the image quality for many equations is too low, for example: line 214, 321 324, etc.

In Fig. S4, it looks like that all the wavelengths (808 nm, 980 nm, 1532 nm) can excite the red emission at 650 nm, which may induce unwanted crosstalk. The authors should comment on the possible reason for this excitation and how to further optimize the UCNPs.

Fig. S13 has very low quality, better images should be provided. “No significant cell apoptosis was detected around the injection site.” To further validate this claim, results of immunostaining like GFAP and IBA1 should be added.

As for the temperature measurements in Figs. S19 and S20, it is more critical to investigate the heating effects within the brain tissue. Also, it is suggested that a thermal model is established to

understand these results. It seems that the temperature rise is about 3-4 degree C, which is too high. It is known that heating over 1-2 degree C may affect the brain activity, even if the tissue is not permanently damaged. (see: <https://www.nature.com/articles/s41593-019-0422-3>)

Optogenetic experiments in Figs. 4 and 5 are performed with a very low laser pulse frequency (0.05-0.1 Hz). Commonly optogenetic excitation requires a much higher frequency (10-100 Hz). Please state the rationale for selecting such low frequencies for stimulation.

In vivo demonstrations are performed with head-fixed mice. The authors should comment on how to apply these techniques on freely moving animals.

Line 41-45 in the supplement, the units are missing.

In the caption for Fig. S6, Line 303-311, the descriptions for Fig. S6e and S6f are missing.

To summarize, the paper is very interesting and should have general interests in the communities of optical materials, bioengineering and neuroscience. To further improve the quality of the paper for broader impacts, I suggest the paper be accepted after addressing my comments above.

Reviewer #2:

Remarks to the Author:

The combination of UCNP and optogenetics have attracted interest from multidisciplinary research communities because of the great tissue penetration depth of NIR light involved for excitation of upconversion processes. This paper entitled "Near-infrared Manipulation of Multiple Neuronal Populations via Trichromatic Upconversion" presents an advancement in UCNP-based neuroregulation by using trichromatic upconversion integrated into a single nanoparticle. This work may attract interests from multidisciplinary research communities, e.g., photonics, material sciences and biomedicine.

Comments:

1. For nanoparticle synthesis, Figure S1a showed an irregular growth (CS4), corresponding to NaGdF₄:50%Nd,10%Yb layer. It is strange that (1) the nanoparticles become smooth when an additional shell layer was applied (CS5), and (2) the size of CS5 nanoparticles seemed to be even smaller than that of CS4 nanoparticles, despite a statistical chart showing their size distribution was presented by the authors. Furthermore, why such abnormal growth doesn't happen in the case in Figure S9. The authors are requested to explain this phenomenon and evaluate the consequence.

2. The component of blue-emitting layer for the trichromatic UCNP is determined to be NaGdF₄:80%Yb,1%Tm by authors. Why not NaYbF₄:1%Tm? Intuitively such a component seems to be more efficient for blocking 980 nm excitation photons.

3. For comparison of trichromatic UCNP and mixture of monochromatic UCNP in Figure 1b, why the red emission from monochromatic UCNP was much stronger than that from trichromatic UCNP. Also, the emission spectra for Figure 1b (bottom panel) are expected to present a direct comparison of the emission properties.

4. The emission spectra in Figure 3a are incomplete, which only showed an optical range of ~570 nm to 700 nm. A full range (400-700 nm) is suggested to present the whole spectra from Er³⁺. Such case is also applicable to Figure S6. In addition, the emission intensity from Er³⁺ basically keeps constant as Tm³⁺ increases from 1% to 10% in Figure 3b, which is inconsistent with the discussion of blocking effect raised in the main text. In Figure 3d, how to measure 980 nm decay under a 980 nm excitation? Better to show the tick and value in the y-coordinate and full spectrum range in the x-coordinate in Figure 3d.

5. Can the authors provide quantitative calculations on the blocking effect of the Nd/Yb layer? It's hard to believe that such a thin shell (a few nanometers) can block most of the excitation light, especially when one considers the very low extinction coefficient of lanthanide ions.

6. When the mixture of AAV-DIO-ChR2-186 mCherry, AAV-CaMKII α -C1V1-EYFP and AAV-fDIO-ChrimsonR-mCherry viruses was stereotaxically injected into the visual cortex (V1) of SOM-Flp:PV-Cre mice, how ChR2, C1V1 and ChrimsonR were selectively expressed in PV, CaMKII α and SOM neurons? The authors tested the neuronal responses two weeks after viruses and UCNP were injected into M2 of PV-Cre mice (Page 13, line 261), so what is the amount of protein expressed in the corresponding neurons during the two weeks and what is the amount of UCNP remained in the M2 after two weeks? Mass analysis should be added.

7. As the green and red emissions contribute simultaneously to promotion on average running distance, any mutual interference for 808 nm excitation? Because the 808 nm excitation could

produce both green and red emission, despite the red emission is weaker than green emission?

8. How were the UCNPs treated (surface modification) before the animal experiment?

9. As the authors showed an opposite relation by 980 nm/808 nm excitations for locomotion modulation, what is the result using 980 nm + 808 nm + 1532nm triplex excitation?

10. The animal study needs to be improved by unfixing the mouse head. Otherwise, I cannot see the necessity and advantage of using UCNPs and NIR irradiation.

Reviewer #3:

Remarks to the Author:

General Comments

Near-infrared (NIR) excited upconversion nanoparticles (UCNPs) provide a convenient approach for activating actuators non-invasively. But an issue with the technology has been the activation of multiple wavelengths selectively. Here the authors present an interesting approach to photo-modulation using UCNP that are spectrally tuned to three wavelengths.

But the excitation-responsive upconversion luminescence is still limited to only two colours. They illustrate in vitro controls combined with in vivo approaches to provide a proof of concept for their UCNPs. This is an exciting and important study that will add new methods for multiple manipulations of neural circuits. Extensive supplementary information is provided regarding the fabrication of the UCNP. Statistics are appropriate.

Major Comments

Minor Comments

There are quite a few spelling and grammatical errors throughout the text.

Abstract. Does not adequately describe the in vivo experiments.

Page 2. Rephrase this sentence 'The multi-colour optogenetics is competent in the independent control of different cell types in the same preparation'.

Page 9: 170 "these tests"

Page 10 186: 'advantage' replace with 'utility'

Page 11. Line 224. The values of temperature should include the starting temperature or a delta value.

Page 11. It is not clear from the text why heating below the surface is not an important factor.

Page 13. Line 254. What does relatively deep anesthesia mean?

Page 17. Line 324. Overheat to overheating

Page 17. Lne 342. Hypothslamus. Spelling error.

Page 17. Line 342. Rewrite this section. I was expecting a discussion of basic circuits in general - that is what new types of experiments are possible. Then a discussion of clinical possibilities with NIR and UCNP.

Page 19. Line 369. Was the animal induced before placing on the stereotaxic apparatus?

Page 19. Line 371. Tuned → turned

Page 19. Line 371. I note that anesthesia was used at 1.5% then reduced to 0.5%. But it seems

as if a surgical plane of anesthesia can't be maintained at this level. To what extent could the animal have been conscious, without a period of recovery? Were any analgesics employed, such as lidocaine, around the craniotomy?

Page 19. Line 380. Throughout whole recordings.

Point-to-Point response to reviewers' comments:

Reviewer #1 (Remarks to the Author):

Optogenetic technology has become a powerful toolset for manipulating brain functions, by targeting on genetically modified neurons expressing light sensitive and wavelength specific opsins. Although blue light sensitive ChR2 is widely used, it is highly demanded to develop a palette of opsins with different excitation spectra, so that different groups of neurons can be separately labeled (*Nature. Methods* 11, 338-346 (2014)). Meanwhile, advanced optical technologies are needed to deliver light at corresponding wavelengths, to selectively interact with these opsins and excite or inhibit these neurons and precisely control neural activities. In this paper, a new type of upconversion nanoparticles (UCNPs) is developed to realize trichromatic emission with infrared illumination. By carefully controlled chemical synthesis, the UCNPs are constructed with multiple coatings, and the energy transfer mechanism determines that the emission colors (blue, green or red) can be tuned dependent on the irradiation wavelengths (980 nm, 808 nm and 1532 nm). These UCNPs are injected into the rodent brains with co-expressed ChR2, C1V1 and ChrimsonR opsins targeting on different neuron types. The authors demonstrate that the manipulations of multiple neuronal populations can be achieved by varying the infrared irradiation wavelengths. In terms of materials synthesis and the unique optical properties of the UCNPs, the results are beautiful and very intriguing. However, my major concerns are mostly in the biological experiments. Detailed comments are listed below:

1. -The spectral responses of ChR2, C1V1 and ChrimsonR are plotted below, with data extracted from: Klapoetke, N. C. et al. Independent optical excitation of distinct neural populations. *Nat. Methods* 11, 338-346 (2014).

Although the paper demonstrate three different emission colors (475 nm, 540 nm and 650 nm) can be excited by different infrared sources (980 nm, 808 nm and 1532 nm) using the designed structure of UCNPs. However, the inherent spectral overlap of ChR2, C1V1 and ChrimsonR cannot be bypassed. It is clear that 475 nm emission can excite both ChR2 and C1V1 (response ratio ~ 1:0.7), and 540 nm emission can excite both C1V1 and ChrimsonR (response ratio ~ 1:0.7), very effectively. It is difficult to believe that these three opsins can be distinctively resolved using the tri-color emission.

Author's reply:

Thanks for your comment. First of all, we agree that the inherent spectral overlap of these three opsins is an important issue in multi-color optogenetics. Previous studies have shown that using designed range of light intensities could minimize the crosstalk between different opsins. For example, in *Nat. Methods* 11, 338-346 (2014), as reviewer mentioned, has discussed about defining an operational range of light intensities to prevent crosstalk in the excitation of two opsins (Chrimson and Chronos), given the overlap between their excitation spectra (Figure 1a in this reply letter). Red light elicited spikes in Chrimson-expressing but not Chronos-expressing cells (Figure 1b in this reply letter). Blue light elicited spikes in Chronos-expressing cells at light intensities as low as 0.05 mW/mm^2 , but Chrimson-expressing neurons were activated by blue light with intensities above 0.5 mW/mm^2 (Figure 1c in this reply letter). These results showed that over an operational range between $0.2\text{-}0.5 \text{ mW/mm}^2$ in blue irradiance, Chronos-expressing neurons can be excited with high fidelity with very low Chrimson-expressing neuron spike probability. With a spectral peak at 590 nm, Chrimson is more red-shifted than other channelrhodopsin (Fig 1d in this reply letter).

Second, we would like to bring into the reviewer's attention that the activation spectra of C1V1 (we used C1V1(T/T)) in the reviewer's figure (maybe VChR1?) are slightly different from the one in the references (e.g. Ref 2; Figure 1e-f in this reply letter). Hence the response ratio of ChR2 and C1V1(T/T) to 475 nm emission was roughly 1: 0.5. To address the reviewer's concern regarding the coactivation of ChR2-expressing and C1V1(T/T)-expressing neurons with 470 nm emission, we injected AAV-DIO-ChR2-mCherry, AAV-CaMKII α -C1V1(T/T)-EYFP, AAV-fDIO-ChrimsonR-mCherry into M2 area of PV-Cre, wild-type and SOM-Flp mice respectively. Three weeks later, we recorded neuronal responses of ChR2-, C1V1(T/T)- and ChrimsonR-expressing cells to trichromatic emission with varying power densities using patch clamp in brain slices (Figure 2a in this reply letter). Figure 2b in this reply letter showed raw traces of responses to 655-nm emission of an example ChrimsonR-expressing cell. We found that when applying 470-nm light with power densities at the range of $0.0015\text{-}0.0025 \text{ mW/mm}^2$, ChR2-expressing cells spiked more frequently than C1V1(T/T)-expressing cells ($\sim 0.8\text{:}0.5$) (Figure 2c in this reply letter). Figure 2d in this reply letter showed two example cells with C1V1(T/T) or ChR2 expression respectively under 470 nm excitation.

We then addressed the reviewer's concern regarding the coactivation of ChrimsonR and C1V1(T/T) neurons with 546 nm emission. 546 nm emission elicited more spikes in C1V1(T/T)-expressing cells than ChrimsonR-expressing cells ($\sim 0.8\text{:}0.4$) at power densities of $0.0005\text{-}0.0015 \text{ mW/mm}^2$ (Figure 2e in this reply letter). Figure 2f in this reply letter showed two example cells with C1V1(T/T) or ChrimsonR expression under 546 nm emission. By comparing our results with those in *Nature Methods* 2014, we found that the response ratios of C1V1(T/T)/ChR2 and C1V1(T/T)/ChrimsonR in our results are larger than that of Chronos/ChrimsonR, despite the fact that the spectral overlap is similar. Note that both Chronos and ChrimsonR were expressed in the same cell in *Nature Methods* 2014, while C1V1(T/T) and ChR2 were expressed in

different types of cells in our case. Figure 2 in this reply letter was added to the revised supplementary information as a new Supplementary Figure 19.

Figure 1 in this reply letter (Figures from Ref 1 and 2, details below). (a-d) Tuning light intensities to prevent crosstalk in the excitation of Chrimson and Chronos (cited from **Figs. 1,4 Ref 1**). (e) Representative traces and summary plot of channel closure time constant (τ_{off}) in cultured neurons expressing the indicated channelrhodopsins; traces are normalized to peak current (cited from **Fig. 4 Ref 2**). (f) Action spectra collected for the indicated channelrhodopsins (colour code as in e). Photocurrents were collected with 2 ms light pulses in HEK293 cells. (cited from **Fig. 4 Ref 2**).

Figure 2 in this reply letter, Supplementary Figure 19 in revised manuscript. Visible light excitation of neural populations in mouse M2 brain slice. (a) Representative of phase contrast (left) and mCherry (right) fluorescent images of ChrimsonR-mCherry fusion transfected neurons. Scale bar, 100 μm . **(b)** Responses of ChrimsonR-expressing cell to 665 nm emission. (top: 0.001585 mW/mm^2 , middle: 0.002913 mW/mm^2 , bottom: 0.006541 mW/mm^2). **(c)** Normalized responses of C1V1 (T/T)-expressing cells ($n = 6$) and ChR2-expressing cells to 470 nm emission ($n = 3$). **(d)** Example of ChR2-(C1V1-) expressing cell to 470 nm emission. **(e)** Normalized responses of C1V1-expressing cells ($n = 9$) and ChrimsonR-expressing cells to 546 nm emission ($n = 5$). **(f)** Example of C1V1-(ChrimsonR-)expressing cell to 546 nm emission.

References:

[1] Klapoetke N C. et al. Independent optical excitation of distinct neural populations. *Nat Methods*, 2014, 11(3): 338-46. doi: 10.1038/nmeth.2836.

[2] Yizhar O, Fenno LE, Prigge M, et al. Neocortical excitation/inhibition balance in information processing and social dysfunction. *Nature*. 2011;477(7363):171-178. Published 2011 Jul 27. doi:10.1038/nature10360

Line 206-210 in revised manuscript

The selectivity of visible light was verified by *in vitro* patch clamp. It is confirmed that ChR2-expressing cells elicited more spikes than C1V1(T/T)-expressing cells (~0.8:0.5) under light of 470-nm. Similarly, light of 546-nm elicited more spikes in C1V1(T/T)-expressing cells than ChrimsonR-expressing cells (~0.8:0.4).

2. To validate the concept for multi-color modulation, the neuronal activities should be recorded at the cellular level with a patch clamp, using cultured neurons or brain slices. For example, co-culture C1V1-expressing cells with the UCNP, then demonstrate that the cells can only be excited by 808 nm, but not by 980 nm or 1532 nm. The same for ChR2- and ChrimsonR expressing cells.

Author's reply:

Thanks a lot for your comment. We injected AAV-DIO-ChR2-mCherry, AAV-CaMKII α -C1V1(T/T)-EYFP and AAV-fDIO-ChrimsonR-mCherry viruses into M2 area of PV-Cre, wild-type and SOM-Flp mice respectively and recorded neuronal activities with patch clamp in brain slices. In the patch clamp experiments, we perfused the brain slices with artificial cerebrospinal fluid (ACSF). The UCNP were diffused in the solution, hence couldn't maintain a stable concentration in the recording chamber as those in *in vivo* experiments. UCNP illuminant were thus manufactured in the form of transparent solid film. Specifically, 1 mL UCNP stock solution were re-dispersed to 4 mL chloroform with 0.40 g dissolved polymethylpropanamide (PMMA). The solid UCNP film was then obtained by dripping solution onto the cover glass and completely evaporating the chloroform solvent (Figure 3a in this reply letter). The brain slice was placed onto the UCNP film and the estimated distance between the film and recorded neurons in the brain slice was roughly 0.25 mm, suggesting that NIR light penetrated through the 0.25 mm thick brain tissue to reach the UCNP film, was converted to visible light and penetrated backwards through the 0.25 mm thick tissue to activate the neurons. In contrast, UCNP in *in vivo* experiments were right next to the recorded neurons. Hence the UCNP-mediated visible light intensity in slice experiments is much lower than that in *in vivo* experiments. We recorded the evoked membrane potentials instead of action

potentials in *in vitro* patch clamp experiments. The results showed that 808 nm NIR light depolarized C1V1(T/T)-expressing cells at 100 - 200 mW/mm² without activating ChR2-expressing cells (Figure 3b in the reply letter). 980 nm NIR light at 50 - 100 mW/mm² excited ChR2-expressing cells while causing little membrane potential elevation in C1V1(T/T)-expressing cells (Figure 3c in the reply letter). These results, together with the figures were included into the revised SI and new Supplementary Figure 20.

Meanwhile, we found that 1532 nm light was largely absorbed by ACSF as reported in Ref3 (Figure 3d, in this reply letter). To evaluate such water absorption effect, we measured transmitted power of NIR lasers through ACSF with different thickness (Table 1 in this reply letter). In the recording chamber, the distance between the brain slice and optical fiber output was about 2 mm. At which depth, 98.8% of 1532 nm light was absorbed by water. In contrast, the absorption of 1532 nm through 400 μm brain tissue was similar to that of 980-nm and 808-nm as shown in Supplementary Figure 16 in the previous manuscript (Supplementary Figure 22 in the revised SI). Therefore, we did not conduct the brain slice experiments with 1532 nm excitation.

Figure 3 in this reply letter. Subthreshold responses of ChR2- and C1V1-expressing cells to NIR lasers. (a) Image of transparent solid UCNP film. **(b)** Subthreshold responses of ChR2- and C1V1-expressing neurons to 808 nm emission (n = 3). **(c)** Subthreshold responses of ChR2- and C1V1-expressing neurons to 980 nm emission (n = 3). **(d)** Absorption spectrum of water through a 1-mm-long path.

OD, optical density (Cited from **Ref. 3 Fig.1d**).

Thickness of ACSF (mm)	Average Detected Power (mw)		
	808-nm	980-nm	1532-nm
0	511.0	481.0	527.2
2	518.1	450.2	4.8
3	519.2	439.3	6.7

Table 1 in this reply letter. Detected powers of NIR lasers transmitted through ACSF with different thickness. The comparative lower power of 808-nm laser at 0-3 mm was caused by reading fluctuation of detector.

Reference:

[3] Hong G, Antaris AL, Dai H. Near-infrared fluorophores for biomedical imaging. *Nat Biomed Eng* 2017, **1**(1): 0010. doi.10.1038/s41551-016-0010

Line 210-212 in revised manuscript

We then confirmed the selectivity to NIR laser powers. ChR2-expressing cells and C1V1(T/T)-expressing cells can be activated by 980-nm and 808-nm respectively (Supplementary Fig. 20).

3. *Fig. S5 shows that the UCNPs have stable PL output for up to 120 mins in solutions. This is not enough. The UCNPs are injected into the animal brain for more than a week (Fig. S13). How to validate that these UCNPs are stable in the brain for such a long time?*

Author's reply:

We thank the reviewer for this constructive comment. The 120 min PL output in previous Fig. S13 was measured under a continuous laser excitation in order to test the photostability. For the stability of the nanoparticles in the physiological environment, it is very difficult to directly test the spectrum of UCNPs with such super small volume (414 nL injection). Instead, we tested the optical spectrum of UCNPs in ACSF stock solution after a week to prove the stability of the nanoparticles in the physiological environment. The mass analyzation was also introduced to evaluate the amount of remaining UCNPs in the brain three weeks after injection. The PL output for UCNPs remains almost constant after one week in ACSF (Figure 4 in this reply letter). For mass analyzation, we injected 414 nL UCNPs into M2 of two wild-type mice. The mice brains of injection sites were dissected after perfusion with saline solution. One of the two brains was dissected the day we injected UCNPs and the other one was dissected three weeks after injection. The brains were calcinated in a furnace at 800 °C for 5 hours. In this process, the organic tissue would be oxidated and volatile, leaving the inorganic lanthanide elements. We measured the residual

content of the rare earth element by ICP-AES elemental analysis. Lanthanide concentration in the brain dissected on the injection day was 1.87 mg/L, while the lanthanide concentration in the other brain dissected three weeks after injection was 1.44 mg/L. These results indicated that at least 77% UCNPs were left after injection for three weeks. Beside mass analyzation, we noticed that within the 40 days of behavioral experiments, the effect was similar in the 1st, 3rd and 5th week upon NIR illumination (Figure 4d in this reply letter), which also indicates that the UCNPs-mediated visible light emission was stable. Each column represents a single trial from one mouse.

Figure 4 in this reply letter, Supplementary Figure 17 in revised manuscript. Physiological stability of UCNPs. (a-c) Emission spectrum of UCNPs under 808-nm, 980-nm and 1532-nm excitation before and after stocked in ACSF for a week. (d) Example behavioral performances of mice at 1st, 3rd and 5th week after UCNP injection. Each column represents a single trial of one mouse. And only one trial of each mouse was recorded one day.

Line 205-208 in revised manuscript

The UCNPs also showed stable photoluminescence output in physiological environment for over a week (Supplementary Fig. 17). Tested by elemental analysis, the mass of UCNPs maintained about 77% after injecting into brain for three weeks.

4. Electron micrographs should be provided, to see the distributions of these UCNPs within the brain tissue. Do they spread outside the cells, or are they uptaken by the cells?

Author's reply:

Thanks a lot for your suggestion. We provided the electron micrograph as shown below to investigate the distribution of the UCNPs within the brain tissue. The results showed that the majority of UCNPs was distributed in extra-cellular space and a small fraction of UCNPs were uptaken by microglia (Figure 5 in this reply letter), which is consistent with the results reported by Chen et.al (Ref. 4 in this reply letter Fig.2B). Figure 5 in the reply letter was included as the new supplementary figure 16 in the revised manuscript.

Figure 5 in this reply letter, Supplementary Figure 16 in revised manuscript. Electron micrographs of UCNPs in M2. Red arrows indicate UCNPs.

Line 198 - 200 in revised manuscript

The distribution of UCNPs was observed by HRTEM, suggesting the majority were distributed in extra-cellular space around the injection site and a small fraction of which may be uptaken by microglia (Supplementary Fig. 16).

Reference:

[4] Chen S, Weitemier AZ, Zeng X, He L, Wang X, Tao Y, *et al.* Near-infrared deep brain stimulation via upconversion nanoparticle-mediated optogenetics. *Science* 2018, **359**(6376): 679-684. doi: 10.1126/science.aaq1144

5. Fig. 4c has very low quality. It is highly recommended that the authors prepare a better brain slice to show more neurons with fluorescence labels, in order to show that three different opsins (ChR2, C1V1 and ChrimsonR) are really expressing in three distinct neuron types, with minimal co-expressions

Author's reply:

Thanks for your suggestion. We injected the mixture of AAV-DIO-ChR2-mCherry, AAV-CaMKII α -C1V1(T/T)-EYFP and AAV-fDIO-ChrimsonR-mCherry viruses into PV-Cre:SOM-Flp mice. Three weeks later, mice were perfused with PBS and 4% PFA (see Methods). GFP labels CaMKII α neurons. mCherry colocalized with PV represented PV neurons whereas mCherry alone represented SOM neurons. The new immunohistochemistry staining results are shown in Figure 6 in this reply letter and included in Figure 4 in revised manuscript.

Figure 6 in this reply letter (Figure 4c in revised manuscript). Expression of C1V1-EYFP, ChrimsonR-mCherry and ChR2-mCherry in CaMKII α , PV and SOM neurons. PV+ neurons were labeled with PV-antibodies to distinguish ChR2-expressing PV neurons from ChrimsonR-expressing SOM neurons. White arrows indicate EYFP-expressing CaMKII α neurons, light blue arrows indicate mCherry-expressing SOM neurons, and pink arrows indicate mCherry-expressing PV neurons. Scale bar, 100 μ m.

6. Fig. 4d shows the distributions of UCNPs in the brain tissue. Unlike viruses that can target on specific cell types or regions, these UCNPs do not have any cell specificity. The natural diffusion of UCNPs should result in an isotropic distribution. Why do these images present an irregular shape of fluorescence? What limits the diffusion of these UCNPs?

Author's reply:

Thanks for this comment. The shape of the UCNPs distribution may be limited by the vicinity of cell membrane and synaptic clefts as reported by previous study (Ref. 4 in this reply letter). UCNPs were localized in the injection area without extensive diffusion. The majority of which was distributed in extra-cellular spaces. In addition, a small fraction of UCNPs were uptaken by microglia. As seen in Figure 7 in this reply letter (Figure 3B in Ref. 4), the distribution of UCNPs was irregular, similar to our results.

Figure 7 in this reply letter (Figure 3B in Ref. 4). Confocal images showing ChR2-expressing PV interneurons and distribution of UCNPs.

Reference:

[4] Chen S, Weitemier AZ, Zeng X, He L, Wang X, Tao Y, *et al.* Near-infrared deep brain stimulation via upconversion nanoparticle-mediated optogenetics. *Science* 2018, **359**(6376): 679-684. doi: 10.1126/science.aaq1144

7. Line 205 in supplement, “Corresponding converted visible light intensities were 0.40 mW/mm² for blue light, 0.13 mW/mm² for green and 0.02 mW/mm² for red at the recording site around 400 μm below the brain surface and were within the optogenetic selectivity criteria.”. To the reviewer’s knowledge, the typical excitation power density required for optogenetic stimulation is around 1-10 mW/mm². It seems that the power generated by these UCNPs is too low for effective modulation.

Author's reply:

Thanks a lot for your comment. As shown previously in our patch clamp experiments with brain slice, visible lights with low power density (0.000319 mW/mm² for C1V1(T/T), 0.000416 mW/mm² for ChR2 and 0.00207 mW/mm² for ChrimsonR) were capable of eliciting light responses (Figure 8 in this reply letter).

Figure 8 in this reply letter. Responses of ChR2-, C1V1(T/T)- and ChrimsonR-expressing cells to visible light emission. (a) Responses of ChR2-expressing cells to 470 nm emission (n = 3). (b) Responses of C1V1(T/T)-expressing cells to 546 nm emission (n = 6). (c) Responses of ChrimsonR-expressing cells to 665 nm emission (n = 5).

8. In supplement, the image quality for many equations is too low, for example: line 214, 321 324, etc.

Author's reply:

Thanks for this suggestion. We've improved the resolution of all the equation images in the revised manuscript.

9. In Fig. S4, it looks like that all the wavelengths (808 nm, 980 nm, 1532 nm) can excite the red emission at 650 nm, which may induce unwanted crosstalk. The authors should comment on the possible reason for this excitation and how to further optimize the UCNPs.

Author's reply:

We thank the reviewer for this constructive comment. The co-existing green/red bands under 808-nm excitation and blue/red bands under 980-nm are actually the characteristic emission bands respectively from Er^{3+} and Tm^{3+} themselves. The green/red and blue/red ratio of our materials is the same as other reported core-shell UCNPs (Ref 4, 5, Figure 8b-c in this reply letter). Only a core-shell design cannot eliminate such characteristic emissions. Nevertheless, the undesired red emission is relatively low. It should be noted that the y-coordinate in Figure S4 is logarithmic for showing a wide range of emission intensity. It's possible to achieve selective activation by controlling laser powers. We've measured the actual emission power ratio. There is an average ratio of 4.0 for blue/red bands under 980-nm illumination and 3.1 for green/red bands under 808 nm illumination (Figure 9d-e in this reply letter). In view of response ratio between CIV1(T/T) (activated by 546 nm) and ChrimsonR (activated by 665 nm), which is 0.8:0.2 at a power of 10-30 μW (Figure 9f in this reply letter), the red emission under 808 nm illumination would have minimal level of activation in ChrimsonR-expressing neurons.

A further improvement, as absolutely single-band emission, could be realized. Methods like surface dye ligand for emission filtering can be applied, but the material structure and application systems involved dye molecule would be much more complicated (Ref 6 in this reply letter).

Figure 9 in this reply letter. (a) Emission spectra of trichromatic UCNPs in this work. (b) Emission spectra of orthogonal two-color UCNPs in **Ref.5**. (c) Emission spectra of single-band UCNPs under 980-nm or 808-nm excitation in **Ref.4**. (d) Photos of UCNPs emission power measurement set up. (e) Table of measured visible light power under the same laser output to in vivo experiments. (f) Recorded cellular level neuronal activities by patch clamp from brain slices expressed CIV1 and ChrimsonR respectively under different visible light powers ($n = 9$ for CIV1 expressing-cells and $n = 5$ for ChrimsonR expressing-cells).

Reference:

[4] Chen S, Weitemier AZ, Zeng X, He L, Wang X, Tao Y, *et al.* Near-infrared deep brain stimulation via upconversion nanoparticle-mediated optogenetics. *Science* 2018, **359**(6376): 679-684. doi: 10.1126/science.aaq1144

[5] Li X, Guo Z, Zhao T, Lu Y, Zhou L, Zhao D, *et al.* Filtration Shell Mediated Power Density Independent Orthogonal Excitations–Emissions Upconversion Luminescence. *Angew Chem Int Ed* 2016, **55**(7): 2464-2469. doi: 10.1002/anie.201510609

[6] Zhou L, Wang R, Yao C, Li X, Wang C, Zhang X, *et al.* Single-band upconversion nanoprobe for multiplexed simultaneous in situ molecular mapping of cancer biomarkers. *Nat Commun* 2015, **6**: 6938.

10. Fig. S13 has very low quality, better images should be provided. “No significant cell apoptosis was detected around the injection site.” To further validate this claim, results of immunostaining like GFAP and IBA1 should be added.

Author’s reply:

Thank you for your suggestion. We injected 414 nL UCNPs into wild-type mice into three sites (0.5 - 1 mm M/L, 1 - 2 mm A/P, 0.4 mm D/V to bregma suture), same as those in behavioral experiments. We conducted immunohistochemistry staining experiments 1 week and 2 weeks after injection and the expression of GFAP (astrocyte) and IBA1 (microglia) are shown in Figure 10 in this reply letter. No significant aggregation of astrocyte was detected while there was a small amount of microglia aggregation around the injection site. This figure was included as the new Supplementary Figure 15 in the revised supplementary information.

Figure 10 in this reply letter, Supplementary Figure 15 in revised manuscript. Immunohistochemistry staining results of GFAP and IBA1 after UCNP injection. Upper panel, GFAP and IBA1 immunostaining of the injection site from mouse after

one week of UCNPs injection. Lower panel, GFAP and IBA1 immunostaining of the injection site from mouse after two weeks of UCNPs injection. UCNPs were observed under 408 nm illumination. Scale bar, 200 μm .

Line 196 - 198 in revised manuscript:

No significant aggregation of astrocyte was detected while there was a small amount of microglia aggregation around the injection site up to two weeks after UCNPs injection (Supplementary Fig. 15).

11. As for the temperature measurements in Figs. S19 and S20, it is more critical to investigate the heating effects within the brain tissue. Also, it is suggested that a thermal model is established to understand these results. It seems that the temperature rise is about 3-4 degree C, which is too high. It is known that heating over 1-2 degree C may affect the brain activity, even if the tissue is not permanently damaged. (see:<https://www.nature.com/articles/s41593-019-0422-3>)

Author's reply:

Thanks a lot for your comment. We agree that the recently published paper that the reviewer mentioned (it was published right after we submitted our manuscript) has clearly shown that heating over 1 - 2 $^{\circ}\text{C}$ may affect the brain activity as well as behavior. But this effect is specific to cell types (medium spiny neurons in that study) with relatively low spike rates. To make sure that our results are significant and meet the requirement, we conducted new behavioral experiments with temperature rise less than 1 $^{\circ}\text{C}$. We first rechose power intensities of NIR lasers by monitoring the temperature over the skull during the stimulation (0.8 mW/mm^2 808-nm, 1.27 mW/mm^2 980-nm, and 1.27 mW/mm^2 1532-nm NIR light, 0.1 s stimulation with 0.1 s off, 5 Hz, the duration is 120 s in total) via a professional infrared thermal imaging camera. The temperature rise is shown below. The total temperature rise is below 1 $^{\circ}\text{C}$. This figure was added as the new Supplementary Figure 30 in the revised manuscript.

Figure 11 in this reply letter, Supplementary Figure 30 in revised manuscript. Curve of temperature rise during NIR laser stimulation in new behavioral experiments. The power densities of 808-nm, 980-nm and 1532-nm laser are 0.8 mW/mm², 1.27 mW/mm² and 1.27 mW/mm², respectively. Stimulation pattern: 5 Hz, 0.1s on, 0.1s off, 120s in total. Temperatures were measured every 10 seconds. The red bar indicates time period of irradiation and the left and right gray column indicates the starting and ending frames of irradiation.

New behavioral experiments were then conducted. We injected the mixture of AAV-DIO-ChR2-mCherry, AAV-CaMKII α -C1V1(T/T)-EYFP and AAV-fDIO-ChrimsonR-mCherry viruses into M2 area of PV-Cre:SOM-Flp mice. 414 nL UCNPs were injected at each site as those three sites in previous behavioral experiments. Head-fixed mice were placed on a jetball in front of a virtual reality (VR) system and trained to run in a virtual one-dimensional corridor scene. The distances were recorded and calculated under different conditions (no light stimulation, 5 Hz 0.8 mW/mm² 808-nm, 1.27 mW/mm² 980-nm, and 1.27 mW/mm² 1532-nm NIR light stimulation transcranially). The NIR light stimulation was given in a pattern of 0.1 s on and 0.1 s off with 120 s in total duration. The behavioral results are shown in Figure 12 in this reply letter. Single trial of each mouse was recorded each day, and the averaged value of 3 trials was calculated. The results of new behavioral experiments were similar as that in the original manuscript. This figure was included as the new Supplementary Figure 31 in the revised manuscript.

Figure 12 in this reply letter, Supplementary Figure 31 in revised manuscript. Transcranial optogenetic multi-color manipulation of mouse locomotion behavior using UCNPs. (a) Running speed of example trials in one-dimensional corridor in VR system under 0.8 mW/mm² 808-nm, 1.27 mW/mm² 980-nm, 1.27 mW/mm² 1532-nm and trichromatic NIR excitations. **(b)** Delta running distances in one-dimensional corridor in VR system under 0.8 mW/mm² 808-nm, 1.27 mW/mm² 980-nm, 1.27 mW/mm² 1532-nm and trichromatic NIR excitations compared with control (no light stimulation) trials. (n = 4 mice, 3 trials for each mouse).

Line 314 - 322 in revised manuscript:

To verify our results, the heating effect was restricted as low as possible to achieve tri-chromatic optogenetic modulation. The power of NIR lasers were finally chosen as 0.8 mW/mm² for 808-nm, 1.27 mW/mm² for 980-nm and 1.27 mW/mm² for 1532-nm lasers after measuring temperature. By which, the temperature rise was below 1 °C (Supplementary Fig. 30). In locomotion test on one-dimensional corridor scene, similar performances of single light modulation were observed as previous results. When 808-nm, 980-nm and 1532-nm lasers were all illuminated onto the skull, average distance traveled was smaller than control, indicating that activation of ChR2-expressing PV neurons dominated the modulation in this test (Supplementary Fig. 31).

To further illustrate that limiting temperature change at brain surface can make sure the brain tissue will not be overheated, we added a thermal model to show the temperature change from the brain surface to inner site. In general, the temperature change inside the brain region is exponentially lower than the surface. Without taking heat spatial propagation into consideration, this thermal diffusion model proves a rapid descend trend along the radial axis. These analyzations are shown in Figure 13 in this reply letter. Correspondingly, an interpretation is added in the revised supplementary information in Supplementary Figure 23.

We note a numerical analyzation used Monte Carlo simulation reported by Joesph et.al (Ref 7 in this reply letter), which proved the light transport and heat buildup indeed had a rapid decay within brain tissue. Meanwhile, the experimental results reported by Chen et.al (Ref 4 in this reply letter) also suggested that temperature fluctuation at local region caused by NIR irradiation was indeed lower than the brain surface.

Figure 13 in this reply letter. Simulation of heat diffusion in brain tissue. (a) Illustration of planar heat diffusion in brain tissue. **(b)** Simulation result of planar diffusion and cylindrical diffusion. **(c)** Illustration of cylindrical heat diffusion in brain tissue.

Line 628 - 651 in revised supplementary information:

Interpretation of Supplementary Figure 23 in revised supplementary information.

For standard one-dimensional model, the heat source surface is regarded as a surface directly contact with the brain (Figure 13a in this reply letter). In such arrangement, temperature as a function of time t and distance x from the surface can be described with an additional term of heat flow induced by blood perfusion (Ref 8 in this reply letter):

$$c_B \frac{\partial \Delta T}{\partial t} = k_B \frac{\partial^2 \Delta T}{\partial x^2} - \omega c \Delta T$$

where c_B and k_B are the specific heat capacity and thermal conductivity of the brain respectively. ω and c are blood flow rate and perivascular tissue thermal conductivity respectively. ΔT is the temperature difference compared with normal tissue, which is what we're interested in. In steady state, this equation gives an exponential decay solution of distance (Figure 12b orange line in this reply letter):

$$\Delta T(x) = \Delta T_0 e^{-\frac{x}{\sqrt{\frac{k_B}{\omega c}}}} = \Delta T_0 e^{-\frac{x}{\lambda}}$$

where ΔT_0 is the temperature difference at contact surface and the constant λ roughly indicates the average diffusion length of temperature induced by heat.

When considering cylindrical thermal diffusion, the heat spot can be regard as a point source with radius of r_0 (Figure 13c blue line in this reply letter). The function given

in cylindrical coordinates follows the below steady state equation (Ref 9 in this reply letter):

$$\frac{1}{r} \frac{d}{dr} \left(r \frac{d\Delta T}{dr} \right) - \lambda^2 \Delta T = 0$$

where λ is as the same as length constant in planar diffusion, r is the radial distance from the heat source center. The boundary condition is $\Delta T(r_0) = \Delta T_0, \Delta T(\infty) = 0$. The solution of this equation is:

$$\Delta T(r) = \Delta T_0 \frac{K_0\left(\frac{r}{\lambda}\right)}{K_0\left(\frac{r_0}{\lambda}\right)}$$

In which, K_0 is the zero order Bessel function of the second kind. These analysis have been added in the revised manuscript. Both of these models show an exponentially decline of temperature difference as a function of distance from the brain surface. It suggests that limiting the temperature change at brain surface can make sure the local region will not be overheating.

Reference

- [4] Chen S, Weitemier AZ, Zeng X, He L, Wang X, Tao Y, *et al.* Near-infrared deep brain stimulation via upconversion nanoparticle-mediated optogenetics. *Science* 2018, **359**(6376): 679-684. doi: 10.1126/science.aaq1144
- [7] Stujenske Joseph M, Spellman T, Gordon Joshua A. Modeling the Spatiotemporal Dynamics of Light and Heat Propagation for In Vivo Optogenetics. *Cell Reports* 2015, **12**(3): 525-534. doi: 10.1016/j.celrep.2015.06.036
- [8] Kastella KG, Fox JR. The Dynamic Response of Brain Temperature to Localized Heating. *Biophys J* 1971, **11**(6): 521-539. doi: 10.1016/S0006-3495(71)86232-X
- [9] Jafari F, Higgins PD. Thermal modeling in cylindrical coordinates using effective conductivity. *IEEE Trans Ultrason Ferroelectr Freq Control* 1989, **36**(2): 191-196. doi: 10.1109/58.19150

12. Optogenetic experiments in Figs. 4 and 5 are performed with a very low laser pulse frequency (0.05-0.1 Hz). Commonly optogenetic excitation requires a much higher frequency (10-100 Hz). Please state the rationale for selecting such low frequencies for stimulation.

Author's reply:

Thank you for your comment. 0.05 - 0.1 Hz stimulation was used in *in vivo* electrophysiology experiments. As mentioned in the reply to Comment 11, near-infrared light has strong heating effects. Unlike the visible light, NIR light is

mostly absorbed by water in the tissue (Figure 3d in this reply letter), which needs longer convection to release the heat. So that high frequency excitation would induce the heat accumulation and the temperature would continue to rise until heat exchange equilibrium at high temperature.

To show how NIR stimulation frequency affects the temperature accumulation, we separately applied 808-nm, 980-nm and 1532-nm NIR lasers on the skull of a mouse and monitored the real-time temperature change by the professional thermal imaging camera (Figure 14 in this reply letter). The NIR lasers of similar power density (3.3 mW/mm^2 808-nm, 3.6 mW/mm^2 980-nm or 3.4 mW/mm^2 1532-nm NIR laser) were used in this test. The temperature rise under 0.05 Hz pulse is the lowest among the three patterns (0.05 Hz, 10 Hz, 20 Hz). As these three lasers were controlled by Arduino micro-controller, it's difficult to give higher pulse frequency (e.g. 50 Hz, 100 Hz) with accurate pulse-width. Figure 14 in this reply letter was included as new Supplementary Figure 24 in revised manuscript.

Figure 14 in this reply letter, Supplementary Figure 24 in revised manuscript. Curves of temperature rise of stimulation of different frequencies. (a) Curve of temperature rises of stimulation of 808-nm laser of 0.05 Hz, 10 Hz and 20 Hz (120 s in total). (b) Curve of temperature rises of stimulation of 980-nm laser of 0.05 Hz, 10 Hz and 20 Hz (120 s in total). (c) Curve of temperature rises of stimulation of 1532-nm laser of 0.05 Hz, 10 Hz and 20 Hz (120 s in total).

Line 243 - 244 in revised manuscript:

To reduce NIR heating effect, a low laser pulse frequency 0.05 Hz was chosen here for *in vivo* electrophysiology recording (Supplementary Fig. 24).

13. *In vivo* demonstrations are performed with head-fixed mice. The authors should comment on how to apply these techniques on freely moving animals.

Author's reply:

Thanks for your comment. The unfixing modulation can be realized by wearable, implantable device or auto-sighting of lasers (Ref 10 in this reply letter). These

complex systems and equipment can be applied to all the laser stimulation methods upon any wavelength. We have added these options into the discussion.

Reference

[10] Montgomery KL, Yeh AJ, Ho JS, Tsao V, Mohan Iyer S, Grosenick L, *et al.* Wirelessly powered, fully internal optogenetics for brain, spinal and peripheral circuits in mice. *Nat Methods* 2015, **12**: 969-974.

Line 407-408 in the revised manuscript:

Further work can be promoted to wearable, implantable device or auto-sighting of lasers in order to a free and deep controls.

14. Line 41-45 in the supplement, the units are missing.

Author's reply:

We're sorry for confusing reviewers with the vacant character. However, we didn't find the missing units in our version. We guess it is because of a computer character repertoire problem, which make the unit *degree centigrad* missed. We've replaced that character to °C in the revised manuscript.

15. In the caption for Fig. S6, Line 303-311, the descriptions for Fig. S6e and S6f are missing.

Author's reply:

We're sorry for missing Fig. S6 caption. Some data in Figure S6 in the previous version were the same as Figure 3b-d. These redundant data were removed after revision and the missing descriptions of Fig.S6e-f are added.

To summarize, the paper is very interesting and should have general interests in the communities of optical materials, bioengineering and neuroscience. To further improve the quality of the paper for broader impacts, I suggest the paper be accepted after addressing my comments above

Reviewer #2 (Remarks to the Author):

The combination of UCNPs and optogenetics have attracted interest from multidisciplinary research communities because of the great tissue penetration depth of NIR light involved for excitation of upconversion processes. This paper entitled “Near-infrared Manipulation of Multiple Neuronal Populations via Trichromatic Upconversion” presents an advancement in UCNP-based neuroregulation by using trichromatic upconversion integrated into a single nanoparticle. This work may attract interests from multidisciplinary research communities, e.g., photonics, material sciences and biomedicine.

Comments:

1. For nanoparticle synthesis, Figure S1a showed an irregular growth (CS4), corresponding to NaGdF4:50%Nd,10%Yb layer. It is strange that (1) the nanoparticles become smooth when an additional shell layer was applied (CS5),

Author’s reply:

Thank you for the constructive comments of material synthesizing process. Actually, these concerns about structure integrity are inherent characteristics in epitaxial growth, which do not affect the intrinsic optical property. The seemingly “abnormal” growth of Nd layer is originated from precursor concentration and misfit strain in core-shell construction. This phenomenon is very common and has been well investigated in the previous reports (Ref 10, 11, 12 in this reply letter). Specifically, the activation energy in heteroepitaxy is described as:

$$\Delta G^* = \Delta G_0^* \left(1 - \frac{U}{E}\right)^2 \left(1 - \frac{2U}{\Delta u}\right)^{-2}$$

Where ΔG_0^* is activation energy for unstrained nucleation. The misfit strain energy U , bonding energy E and chemical potential Δu determine the activation energy and the growth orientation. Nd^{3+} has a larger ion radius than the others (Nd^{3+} : 0.0995 nm, Yb^{3+} : 0.0858 nm, Y^{3+} : 0.0893 nm, Gd^{3+} : 0.0038 nm) with a lower reactivity. So that coating Nd on core with large size tend to result in a selective growth. The size “smoothing” effect is similar to Ostwald ripening. In the subsequent coating steps, newly added precursor (Gd) with less misfit has relative high diffusion rate than deposition rate and thus homogenize the particle size (Figure 15a in this reply letter). This phenomenon can be seen in the TEM images of final products, in which the edge of CS4 is jagging but that for CS7 is uniform (Figure 15b in this reply letter).

Figure 15 in this reply letter. (a) Schematic illustration for the heterogeneous and homogeneous epitaxy (cite from Ref 12). (b) HRTEM photo of trichromatic nanoparticles with edge profile indication of CS4 and CS7. Scale bar = 100 nm. (c) TEM of CS4 nanoparticles and particle size measurement. Yellow scale bar=100 nm. (d) TEM of CS5 nanoparticles and particle size measurement. Brown scale bar=100 nm.

(2) the size of CS5 nanoparticles seemed to be even smaller than that of CS4 nanoparticles, despite a statistical chart showing their size distribution was presented by the authors.

Author's reply:

We're sorry for confusing reviewers about size comparison of CS4 and CS5. It may be caused by different scale bar presentation. To show the contrast of each layer in all samples (CS1-CS7) clearly, we've cropped the TEM images from different magnification and rearranged the scale bars. A precise indication is shown in Figure 15c-d in this reply letter. Correspondingly, a re-scaled picture of CS4 is presented in revised manuscript (Figure S1).

(3) Furthermore, why such abnormal growth doesn't happen in the case in Figure S9. The authors are requested to explain this phenomenon and evaluate the consequence.

Author's reply:

As discussed above, the heterogeneous growth is ascribed to misfit strain and precursor concentration. On the one hand, different Gd^{3+}/Yb^{3+} ratios on shell and $Gd^{3+}/Yb^{3+}/Nd^{3+}$ in precursor result in diverse radius variances at interface. As what can be seen in Figure S9a, the samples with highest Yb^{3+}/Nd^{3+} also show a heterogeneous coating. On the other hand, the seeds in Figure S9 are individually synthesized, but the samples in Figure S1 come from layer-by-layer growth on a seed of ~22 nm size. Although the sizes of these two samples are similar, the one in Figure S9 has relatively low seed concentration (large core). Thus, a fixed amount of precursor brings about higher precursor/seed ratio and results in a thicker shell without heterogeneous growth. On the contrary, in the samples of Figure S1, lower precursor/seed ratio shows selective aggregation.

Reference

[10] I. V. Markov, *Crystal Growth for Beginners: Fundamentals of Nucleation, Crystal Growth and Epitaxy*, World Scientific, Singapore 1995.

[11] Zhao J, Chen B, Wang F. Shedding Light on the Role of Misfit Strain in Controlling Core-Shell Nanocrystals. *Adv Mater* 2020, **32**(46): e2004142.

[12] Xie S, Choi S-I, Lu N, Roling LT, Herron JA, Zhang L, *et al.* Atomic Layer-by-Layer Deposition of Pt on Pd Nanocubes for Catalysts with Enhanced Activity and Durability toward Oxygen Reduction. *Nano Lett* 2014, **14**(6): 3570-3576.

[13] Wang P, Wang C, Lu L, Li X, Wang W, Zhao M, *et al.* Kinetics-mediate fabrication of multi-model bioimaging lanthanide nanoplates with controllable surface roughness for blood brain barrier transportation. *Biomaterials* 2017, **141**: 223-232. doi: 10.1016/j.biomaterials.2017.06.040

2. The component of blue-emitting layer for the trichromatic UCNP is determined to be $NaGdF_4:80\%Yb,1\%Tm$ by authors. Why not $NaYbF_4:1\%Tm$? Intuitively such a component seems to be more efficient for blocking 980 nm excitation photons.

Author's reply:

We thank for this constructive comment. The reason why we chose the value of 80% is in consideration of balancing the blue emission intensity and dissipating efficiency. Although the concentration over 80% may show some increase of dissipation, the $NaYbF_4$ shell actually results in relatively low emission intensity. We indeed found such concentration effect in preliminary experiments before, on a simpler structure $NaYF_4@NaYF_4:x\%Yb,1\%Tm@NaYF_4$. It shows that when Yb concentration is increased over 80% to 99%, shell thickness and emission intensity are decreased (Figure 16 a-c in this reply letter), which goes against an effective emission for application. Correspondingly, this explanation is added in Figure S7 in revised supplementary information.

The reasons may be attributed to the short of matrix. On the one hand, sensitization is weakened without phonon-assistance of matrix (Gd/Y), on the other hand, there is an inevitable leakage of Yb to the outer layer surface when coating subsequent shells on NaYbF₄ (Ref 12 in this reply letter). It forms a quenching pathway and attenuates the upconversion intensity (Figure 16a in this reply letter, cited from Ref 11).

Figure 16 in this reply letter. (a) Illustration of high concentration inner elements leakage via high temperature long time reaction (cite from Ref 11 Figure 1). (b) Luminescence spectra of UCNPs NaYF₄@NaYF₄: x%Yb,1%Tm@NaYF₄ (x = 20, 50, 80 99). (c) Integrating Intensity of 450-nm and 475-nm bands as a function of Yb doping concentration in b. (d) TEM images of UCNPs NaYF₄@NaYF₄: x%Yb,1%Tm, which derived from the same core. Scale bar=50 nm.

Reference

[12] Liu L, Li X, Fan Y, Wang C, El-Toni AM, Alhoshan MS, *et al.* Elemental Migration in Core/Shell Structured Lanthanide Doped Nanoparticles. *Chem Mater* 2019, **31**(15): 5608-5615.

3. For comparison of trichromatic UCNPs and mixture of monochromatic UCNPs in Figure 1b, why the red emission from monochromatic UCNPs was much stronger than that from trichromatic UCNPs. Also, the emission spectra for Figure 1b (bottom panel) are expected to present a direct comparison of the emission properties.)

Author's reply:

Actually, the photos of trichromatic UCNPs emission and mixture samples' emission are simultaneously captured in one picture by direct irradiation at laser

output from the left side. Due to a larger divergence angle of fiber-coupled interface of 1532-nm laser (Figure 17a in this reply letter), the laser beam is spread out and power density at left side is higher than the right side result in a variance in excitation power density. In contrast, 808-nm and 980-nm lasers are free-space outputs with straight beam (Figure 17a in this reply letter), which do not show such large divergence. We've re-captured these samples individually and replaced them (Figure 1 in revised manuscript, Figure 17b in this reply letter).

The actual emission spectra, which are derived from spectroscopy by collimating lights with the same excitation powers, were already shown at Figure 4a in previous manuscript, along with the active spectra of three optogenetic proteins (Figure 17c in this reply letter).

Figure 17 in this reply letter. (a) Photos of 808-nm, 980-nm and 1532-nm laser outputs. (b) Photos of trichromatic UCNPs and mixture of three types under 980-nm, 808-nm or 1532-nm excitation respectively. (c) Emission spectra of trichromatic UCNPs (colored full line) and mixture of three monochromatic UCNPs (black full line) under 808, 980 and 1532 nm NIR and activated spectra of three optogenetic proteins (dash line) (Figure 4a in manuscript).

4. The emission spectra in Figure 3a are incomplete, which only showed an optical range of ~570 nm to 700 nm. A full range (400-700 nm) is suggested to present the whole spectra from Er³⁺. Such case is also applicable to Figure S6. In addition, the emission intensity from Er³⁺ basically keeps constant as Tm³⁺ increases from 1% to 10% in Figure 3b, which is inconsistent with the discussion of blocking effect raised in the main text. In Figure 3d, how to measure 980 nm decay under a 980 nm excitation? Better to show the tick and value in the y-coordinate and full spectrum range in the x-coordinate in Figure 3d.

Author's reply:

(1) We thank for these revising advices about data presentation. The spectra and lifetime with a full range are shown in the revised manuscript (Figure 18 b,d in this reply letter and Figure 3a,d in revised manuscript). Some data in Figure S6 in previous manuscript is the same as Figure 3b-d in main text. These redundant data were removed after revision.

Figure 18 in this reply letter. (a-b) Emission spectra of $\text{NaErF}_4@NaYF_4@NaGdF_4:x\%Yb@NaYF_4$ UCNP (x = 0,10,49,80) with different shell composition before (a) and after revision (b). (c-d) Luminescence decay curves at 980 nm of the obtained UCNP with different Tm^{3+} doping ratio under 980 nm excitation before (c) and after revision (d).

In addition, the emission intensity from Er^{3+} basically keeps constant as Tm^{3+} increases from 1% to 10% in Figure 3b, which is inconsistent with the discussion of blocking effect raised in the main text.

Author's reply:

(2) The further 5% and 10% doping of Tm indeed decrease the red intensity from ~56.4% to ~67.1%, although they do not show remarkable change as compared with the change of 1% Tm doping. It indicates that 1% of Tm is sufficient to de-saturate the activate population of Yb under continuous excitation and Tm can also absorb 980-nm photons by itself. We think it is consistent with the discussion in line 135 in

previous manuscript: “...**even only 1 % of Tm^{3+} activator co-doped with 49 % Yb^{3+} sensitizer results in highly enhanced red emission suppression**”.

In Figure 3d, how to measure 980 nm decay under a 980 nm excitation? Better to show the tick and value in the y-coordinate and full spectrum range in the x-coordinate in Figure 3d.

Author’s reply:

(3). Measuring the 980 nm decay under the same wavelength is as the same as the other lifetime measurements. We guess that reviewer may concern about the laser interference. It actually disturbed only the steady state measurement (spectrum). But the transient emission curve (lifetime) is obtained by time-correlated single photon counting, in which photons were recorded at the time after ceasing laser signal. This is feasible and shows a high quantum efficiency (Ref 13 in this reply letter). Corresponding to the above comments, data with full range of x and y-coordinate is shown in revised manuscript (Figure 18 d in this reply letter and Figure 3d in revised manuscript).

Reference

[13] Gu Y, Guo Z, Yuan W, Kong M, Liu Y, Liu Y, *et al.* High-sensitivity imaging of time-domain near-infrared light transducer. *Nat Photon* 2019, **13**(8): 525-531.

5. Can the authors provide quantitative calculations on the blocking effect of the Nd/Yb layer? It’s hard to believe that such a thin shell (a few nanometers) can block most of the excitation light, especially when one considers the very low extinction coefficient of lanthanide ions.

Author’s reply:

We considered the dissipation effect is similar to the competitive-absorption-process, which follows the law of absorption. The emission intensity change can be described as:

$$I(z) = I_0 e^{-mzS_\lambda}$$

Where I_0 and $I(z)$ are the intensity before and after absorption. S_λ is the absorption cross-section at specific wavelength. z is the optical path and m is the absorbent volume concentration. In core-shell structure, m and z is corresponding to the doping amount and shell thickness of outer layer. For 980-nm dissipating, it is related to S_{980} of Yb and Er. For 808-nm dissipating, S_{808} of Nd and Er is critical (Figure 19a in this reply letter). The absorbance test result of Yb/Er/Nd ions shows that there is 2.4 folds difference of S_{980} for Yb and Er, but is nearly 31 folds difference of S_{808} for Nd and Er (Figure 19b in this reply letter). In our experiments, 80% of Yb doping concentration (m) and ~10 nm shell thickness (z) are managed to

absorb 980-nm photons. In contrast, 50% of Nd doping concentration and 2 nm shell thickness is about 1/8 to the former when compared the product of mz . But the difference of S_λ is 13 times greater than the former. So, it is sufficient to absorb 808-nm photons with such parameters. Note that previous research also suggested that surface dye ligand with larger absorption difference at specific wavelength can better filter the excitation light (Ref 14 in this reply letter, Figure 19c in this reply letter).

Figure 19 in this reply letter. (a) Table summarizing the three cases involved in trichromatic dissipation process. (b) The extinction spectra of lanthanide Yb/Er/Nd in chloride aqueous solutions with the same concentration. (c) The absorption competition effect between dye ligand and lanthanide particles (cite from Ref 14)

Reference

[14] Wang S, Liu L, Fan Y, El-Toni AM, Alhoshan MS, li D, *et al.* In Vivo High-resolution Ratiometric Fluorescence Imaging of Inflammation Using NIR-II Nanoprobes with 1550 nm Emission. *Nano Lett* 2019, **19**(4): 2418-2427

6. When the mixture of AAV-DIO-ChR2-186 mCherry, AAV-CaMKII α -C1V1-EYFP and AAV-fDIO-ChrimsonR-mCherry viruses was stereotaxically injected into the visual cortex (V1) of SOM-Flp: PV-Cre mice, how ChR2, C1V1 and ChrimsonR were selectively expressed in PV, CaMKII α and SOM neurons?

Author's reply:

Thanks for your comment. The selective expression is based on transgenic tools as Cre/LoxP recombination system, FLP/FRT system and the promoter restriction.

(1) ChR2 was selectively expressed in PV neurons with Cre/LoxP recombination system. LoxP sites could be recognized by Cre recombinase, resulting in inversion or excision of target genes (Figure 20a-b in this reply letter). DIO involves two wild-type LoxP sites and two mutant LoxP sites. Two pairs of mismatched LoxP sites could invert and turn on DIO, then one of the LoxP partners would be eliminated to prevent a second inversion (Figure 20c in this reply letter). In our experiment, PV-Cre transgenic mice were utilized, in which Cre recombinase was selectively expressed in PV neurons. By injecting AAV-DIO-ChR2 virus into PV-Cre mice, Cre-recombinase in PV neurons recognized DIO in cells infected by AAV virus and turns on ChR2 expression. In consequence, ChR2 was selectively expressed in PV neurons.

ChrimsonR was expressed in SOM neurons with FLP/FRT system. The FLP-FRT system is similar to the Cre-LoxP system. FLP could recognize a pair of FLP recombinase target (FRT) sequences that flank a target gene. f in fDIO stands for FLP/FRT system.

C1V1 was expressed in CaMKII α neurons by CaMKII α promoter. The CaMKII α promoter makes expression restricted to excitatory neurons.

Figure 20 in this reply letter. Schematics of gene expression with Cre/LoxP and DIO system. (a) Schematics of inversion (b) Schematics of excision (c) Schematics of gene expression with DIO.

The authors tested the neuronal responses two weeks after viruses and UCNPs were injected into M2 of PV-Cre mice (Page 13, line 261), so what is the amount of protein expressed in the corresponding neurons during the two weeks and what is the amount of UCNPs remained in the M2 after two weeks? Mass analysis should be added.

(2) To evaluate the level of optogenetic protein expression, we injected the mixture of of AAV-DIO-ChR2-mCherry, AAV-CaMKII α -C1V1(T/T)-EYFP and AAV-fDIO-ChrimsonR-mCherry viruses into M2 area of PV-Cre:SOM-Flp mice.

After immunohistochemistry staining, we took photos of brain slices (both injected and uninjected sites) under the same exposure time (10 ms for GFP, 30 ms for PV and 3 ms for mCherry) and calculated the fluorescence intensity. The results are shown in Figure 21 in this reply letter. There is a significant difference between injected and uninjected site, which suggest an effective expression of optogenetic proteins in two weeks.

Figure 21 in this reply letter. Fluorescence intensity of brain slices. (a) Example brain slices of injected side (right column) and uninjected side (left column). (b) Fluorescence intensities of GFP and mCherry of both sides (5 slices for each side, 5 areas for each slice; *** $p = 0.0005$, * $p = 0.029$; paired t-test). Scale bar, 100 μm .

(3) To evaluate the amount of UCNPs preservation in two weeks, we made the mass measurement by elemental analysis. We injected 414 nL UCNPs into M2 of two wild-type mice. The mice brains of injection sites were dissected after perfusion with saline solution. One of the two brains was dissected the day we injected UCNPs and the other one was dissected three weeks after injection. The brains were calcinated in a furnace at 800 °C for 5 hours. In this process, the organic tissue would be oxidated and volatile, leaving the inorganic lanthanide elements. We measured the total amount of residues dissolved in aqua regia solution with ICP-AES elemental analysis. The reported lanthanide ions concentration in the brain dissected on the injection day was 1.87 mg/L, while the other brain dissected three weeks after injection was 1.44 mg/L. Given that the brains were pre-perfused by saline solution and there is transfer loss of lanthanide concentration, these results indicated at least over 77% UCNPs were left three weeks after injection. It proves that most of UCNPs are remained after three weeks.

7. As the green and red emissions contribute simultaneously to promotion on average running distance, any mutual interference for 808 nm excitation? Because the 808 nm

excitation could produce both green and red emission, despite the red emission is weaker than green emission?

Author's reply:

Thank you for this comment. Indeed, 808 nm excitation would induce undesired red emission. It is actually the characteristic emission bands from Er^{3+} itself in the shell other than ineffective dissipation. The green/red ratio of our materials is as the same as other reported core-shell UCNPs (Ref 4, 5, Figure 9b-c in this reply letter). A core-shell structure design only cannot eliminate such characteristic emissions. Nevertheless, the undesired red emission is relatively low. We measured the actual emission power ratio and neuron activation thresholds to verify that controlling laser powers is feasible to have a selective color activation. There is an average ratio of 3.1 for green/red bands under 808 nm illumination (Figure 22 b in this reply letter). In view of response ratio between C1V1(T/T) (activated by 546 nm) and ChrimsonR (activated by 665 nm), which is 0.8:0.2 at a power of 10-30 μW (Figure 22c in this reply letter), the red emission under 808 nm illumination would have minimal level of activation in ChrimsonR-expressing neurons. These results were added in the revised manuscript as Supplementary Figure 28.

For *in vivo* test in the original manuscript, 0.70 mW/mm^2 green light emission is accompanied by 0.22 mW/mm^2 red emission as calculated. At such level, there wasn't significant response of ChrimsonR-expressing neurons (Figure 22d in this reply letter). Hence we think the mutual interference from such low red intensity is negligible. That being said, when excitation power density is high enough, the chromatic selectivity under 808-nm will be weakened. However, such upper limit will be much higher (2-3 times) than the thermal limit of NIR application in the *in vivo* experiments. Given that we would not apply a power beyond the thermal limit, we didn't make more investigation for this outside boundary.

Line 229-233 in the revised manuscript:

Note that there is still a relatively low undesired red band under 808-nm. We measured the actual emission power ration of green/red and compared that with the response ratio of C1V1 (activated by 546 nm) and ChrimsonR (activated by 665 nm), which is 0.8:0.2 at a power of 10-30 μW. The result shows that such low red illumination would have minimal level of activation in ChrimsonR-expressing neurons (Supplementary Fig. 28).

8. How were the UCNPs treated (surface modification) before the animal experiment?

Author's reply:

We thank the reviewer for this constructive comment. We're sorry for not clearly providing the nanoparticles surface modification process. The UCNPs were firstly dispersed in cyclohexane after fabrication and transfer to saline solution by a reported filming-rehydration methods (Ref 15 in this reply letter). The phospholipid package provides a hydrophobic inner environment for UCNPs and would not degrade like silica coating. The detailed modification process is added in the Method section in revised manuscript.

Methods in Supplementary Information:

Surface modification of nanoparticles. The synthesized nanoparticles were transferred into liquid phase with the method previously reported by Yao et al. Typically, 1 mL of oleic acid capped UCNPs in cyclohexane (10 mg/mL) was re-dispersed into 1 mL chloroform and mixed with 1 mL chloroform solution containing 12.5 mg DSPE-PEG-OCH₃ in a 250 mL flask. Then the solvent was rotated by 100 rpm and evaporated for 20 min. The resulting mixed film was heated at 80 °C for 5 min to completely remove the solvent. Then film was hydrated by 20 mL water with vigorous sonication. The solution was transferred to a microtube and excess lipids were purified by ultracentrifugation (15000 rpm, 10 min). The subsequent sediment was re-dispersed into 1 mL saline solution.

Reference

[15] Yao C, Wang P, Li X, Hu X, Hou J, Wang L, *et al.* Near-Infrared-Triggered Azobenzene-Liposome/Upconversion Nanoparticle Hybrid Vesicles for Remotely Controlled Drug Delivery to Overcome Cancer Multidrug Resistance. *Adv Mater* 2016, **28**(42): 9341-9348.

9. As the authors showed an opposite relation by 980 nm/808 nm excitations for locomotion modulation, what is the result using 980 nm + 808 nm + 1532nm triplex excitation?

Author's reply:

Thanks for your comment. To address the question, new behavioral experiments were conducted. We injected the mixture of AAV-DIO-ChR2-mCherry, AAV-CaMKII-C1V1(T/T)-EYFP and AAV-fDIO-ChrimsonR-mCherry into M2 area of PV-Cre:SOM-Flp mice. 414 nL UCNPs were also injected at each site as those in previous behavioral experiments. Head-fixed mice were placed on a jetball in front of a virtual reality (VR) system and trained to run in a virtual one-dimensional corridor environment. The distances were recorded and calculated under different conditions (no light stimulation, 5 Hz 0.8 mW/mm² 808-nm, 1.27 mW/mm² 980-nm, and 1.27

mW/mm² 1532-nm NIR light stimulation transcranially). The NIR light stimulation was given in a pattern that 0.1s on and 0.1s off with 120s in total. The behavioral results are shown in Figure 12 in this reply letter. When using 980 nm + 808 nm + 1532 nm triplex excitation, both velocity and distance traveled were smaller comparing to control, indicating that activation of ChR2-expressing PV neurons dominated the modulation in this test. This figure was included as the new Supplementary Figure 31 in the revised manuscript.

Figure 12 in this reply letter, Supplementary Figure 31 in revised manuscript. Transcranial optogenetic multi-color manipulation of mouse locomotion behavior using UCNPs. (a) Running speed of example trials in one-dimensional corridor in VR system under 0.8 mW/mm² 808-nm, 1.27 mW/mm² 980-nm, 1.27 mW/mm² 1532-nm and trichromatic NIR excitations. **(b)** Delta running distances in one-dimensional corridor in VR system under 0.8 mW/mm² 808-nm, 1.27 mW/mm² 980-nm, 1.27 mW/mm² 1532-nm and trichromatic NIR excitations compared with control (no light stimulation) trials. (n = 4 mice, 3 trials for each mouse).

Line 314 - 327 in revised manuscript:

To verify our results, the heating effect was restricted as low as possible to achieve tri-chromatic optogenetic modulation. The power of NIR lasers were finally chosen as 0.8 mW/mm² for 808-nm, 1.27 mW/mm² for 980-nm and 1.27 mW/mm² for 1532-nm lasers after measuring temperature. By which, the temperature rise was below 1 °C

(Supplementary Fig. 30). In locomotion test on one-dimensional corridor scene, similar performances of single light modulation were observed as previous results. When 808-nm, 980-nm and 1532-nm lasers were all illuminated onto the skull, average distance traveled was smaller than control, indicating that activation of ChR2-expressing PV neurons dominated the modulation in this test (Supplementary Fig. 31). The reason for this dominating effect may lie in the difference of NIR power density and conversion efficiency as well as the activation thresholds of ChR2, C1V1 and ChrimsonR, in which PV neurons expressing ChR2 is the most of which being activated. The depolarization time τ_{off} of optogenetic protein may also matter. Besides, there is chance that PV neurons suppress activities globally while SOM and CaMKII α neurons function locally, which may also count for the observed domination.

10. The animal study needs to be improved by unfixing the mouse head. Otherwise, I cannot see the necessity and advantage of using UCNP and NIR irradiation.

Author's reply:

Thanks for this advice. We agree with the reviewer that one major advantage of UCNP and NIR irradiation is in free-moving animals. Previous study has already manufactured some wearable/implantable devices (e.g. Ref 10 in this reply letter) for such application. Auto-sighting of lasers can also be implemented in free-moving schemes. These technological improvements can be readily applied to the laser stimulation methods upon all wavelengths. It may realize both the deep penetration with multi-colors and also a further remote stimulation.

Reference

[10] Montgomery KL, Yeh AJ, Ho JS, Tsao V, Mohan Iyer S, Grosenick L, *et al.* Wirelessly powered, fully internal optogenetics for brain, spinal and peripheral circuits in mice. *Nat Methods* 2015, **12**: 969-974.

Reviewer #3 (Remarks to the Author):

General Comments

Near-infrared (NIR) excited upconversion nanoparticles (UCNPs) provide a convenient approach for activating actuators non-invasively. But an issue with the technology has been the activation of multiple wavelengths selectively. Here the authors present an interesting approach to photo-modulation using UCNP that are spectrally tuned to three wavelengths.

But the excitation-responsive upconversion luminescence is still limited to only two colours. They illustrate in vitro controls combined with in vivo approaches to provide a proof of concept for their UCNPs. This is an exciting and important study that will add new methods for multiple manipulations of neural circuits. Extensive supplementary information is provided regarding the fabrication of the UCNP. Statistics are appropriate.

Major Comments

Minor Comments

There are quite a few spelling and grammatical errors throughout the text.

Abstract. Does not adequately describe the in vivo experiments.

Author's reply:

Thanks for your comment. We have added descriptions of *in vivo* experiments in the abstract of the revised manuscript.

Line 23 - 29 in revised manuscript:

...Such stimulation with tunable intensity not only activated distinct neuronal populations selectively, but also achieved transcranial selective modulation of the running speed of awake-behaving mice in virtual reality that presenting 808 nm or 1532 nm NIR laser resulted in an increased travelling distance while presenting 980 nm NIR laser had an opposite effect. Dual or triple presentation of 808 nm, 980 nm and 1532 nm NIR laser lead to a gradient modulation of the running behavior. Our study opens up a possibility of multi-color upconversion optogenetics.

Page 2. Rephrase this sentence 'The multi-colour optogenetics is competent in the independent control of different cell types in the same preparation' .

Author's reply:

The corresponding sentence is rephrased to '*Independent control of different cell types is workable by multi-color modulation*' .

Page 9: 170 "these tests"

Author's reply:

The corresponding sentence was revised.

Caption of Figure 3 in revised manuscript:

All of the UCNPs samples in these tests were dispersed in cyclohexane

Page 10 186: 'advantage' replace with 'utility'

Author's reply:

The corresponding sentence was revised.

Line 190 in revised manuscript:

To show the utility of our trichromatic UCNPs in in vivo optogenetic experiment...

Page 11. Line 224. The values of temperature should include the starting temperature or a delta value.

Author's reply:

Thanks for your suggestion. The information of delta temperatures was added in the revised manuscript.

Line 246-248 in revised manuscript:

..., it is found that the overall temperature on the brain surface under 980, 808 and 1532 nm illumination increased from 30.61 to 33.28°C ($\Delta^{\circ}\text{C} = 2.67 \pm 0.18^{\circ}\text{C}$), 30.62 to 33.44 °C ($\Delta^{\circ}\text{C} = 2.82 \pm 0.09^{\circ}\text{C}$) and 30.78 to 33.99°C($\Delta^{\circ}\text{C} = 3.21 \pm 0.08^{\circ}\text{C}$) (Supplementary Fig. 25).

Page 11. It is not clear from the text why heating below the surface is not an important factor.

Author's reply:

Thanks for this comment. Heating below the surface is important, we intend to say that limiting temperature change at brain surface can make sure the inner local region will not be overheating. In general, the temperature change at inner region is

exponentially lower than the surface. The corresponding statements were revised.

Line 242-243 in revised manuscript:

So that heating effects were evaluated from brain surface, which has the highest temperature during NIR light irradiation (Supplementary Fig. 23).

Page 13. Line 254. What does relatively deep anesthesia mean?

Author's reply:

Thanks for your comment. We performed *in vivo* electrophysiology recording firstly with 1.5% isoflurane and then kept at 0.5% isoflurane throughout the experiment. Under anesthesia of 1.5% isoflurane, the respiratory rate was about 36 bpm, which increased slightly to 38 bpm upon 0.5% isoflurane.

Page 17. Line 324. Overheat to overheating

Author's reply:

The corresponding sentence was revised.

Line 363-364 in revised manuscript:

We have also carefully measured the safe light intensity for NIR stimulation to prevent overheating in the brain tissue.

Page 17. Lne 342. Hypothslamus. Spelling error.

Author's reply:

We've rewritten the Discussion section and the sentences containing hypothalamus was removed.

Page 17. Line 342. Rewrite this section. I was expecting a discussion of basic circuits in general - that is what new types of experiments are possible. Then a discussion of clinical possibilities with NIR and UCNP.

Author's reply:

Thanks for your suggestion. We've rewritten this section in revised manuscript.

Line 382-405 in revised manuscript:

By elaborate design and detailed testing, we managed to independently modulate multiple neuronal types with NIR lights and UCNPs mediated multi-color optogenetics. One long-term goal of neuroscience is the recording activities of distinct neuronal populations in the highly complicated and holistic brain network, as well as the manipulation of selective neurons to understand what leads to complicated behaviors including vision, emotion, learning and memory. Such task is extremely challenging, given that the anatomical and functional structure of the brain network is often hierarchical and reciprocal, with multiple inputs from different brain regions and multiple outputs to others. When the brain is at work, information was processed within the brain network multidirectionally. Therefore, studying the brain circuits with one optogenetic protein would only allow unidirectional dissection of the behavior. Developing next generation multi-color optogenetic tools with UCNPs will facilitate the multidirectional dissection of complicated brain circuits. Our breakthrough would provide a first-order method for the characterization of hierarchical signal processing streams. Specifically, how inputs with varying levels of coherence from multiple brain regions lead to different emotional outcomes, as well as how activities of ensembles at different hierarchical levels act collectively to contribute to the integration of sensory information.

A recent annual report of Deep Brain Stimulation (DBS) think tank has mentioned advances in optogenetics, which have now emerged as a tool for comprehending neurobiology of diseases and inspired cutting-edge DBS ^[16]. Transcranial optogenetic photoactivation opens up the possibility to explore therapeutic interventions for neurological diseases. With the development of multi-color transcranial optogenetics, we can now target multiple brain areas simultaneously. For instance, bed nucleus of the stria terminalis (BNST) circuit including hypothalamus, parabrachial nucleus and ventral tegmental area is thought to be involved in anxiety^[17]. Coherent manipulation of the BNST circuit will potentially enable the development of precise treatments.

Reference

[16] Vedam-Mai V, Deisseroth K, Giordano J, Lazaro-Munoz G, Chiong W, Suthana N, Langevin JP, Gill J, Goodman W, Provenza NR, Halpern CH, Shivacharan RS, Cunningham TN, Sheth SA, Pouratian N, Scangos KW, Mayberg HS, Horn A, Johnson KA, Butson CR, Gilron R, de Hemptinne C, Wilt R, Yaroshinsky M, Little S, Starr P, Worrell G, Shirvalkar P, Chang E, Volkmann J, Muthuraman M, Groppa S, Kühn AA, Li L, Johnson M, Otto KJ, Raike R, Goetz S, Wu C, Silburn P, Cheeran B, Pathak YJ, Malekmohammadi M, Gunduz A, Wong JK, Cernera S, Wagle Shukla A, Ramirez-Zamora A, Deeb W, Patterson A, Foote KD, Okun MS. Proceedings of the Eighth Annual Deep Brain Stimulation Think Tank: Advances in Optogenetics, Ethical Issues Affecting DBS Research, Neuromodulatory Approaches for Depression, Adaptive Neurostimulation, and Emerging DBS Technologies. *Front Hum Neurosci*. 2021 Apr 19;15:644593. doi: 10.3389/fnhum.2021.644593. PMID: 33953663; PMCID: PMC8092047.

[17] Kim SY, Adhikari A, Lee SY, Marshel JH, Kim CK, Mallory CS, Lo M, Pak S, Mattis J, Lim BK, Malenka RC, Warden MR, Neve R, Tye KM, Deisseroth K. Diverging neural pathways assemble a behavioural state from separable features in anxiety. *Nature*. 2013 Apr 11;496(7444):219-23. doi: 10.1038/nature12018. Epub 2013 Mar 20. PMID: 23515158; PMCID: PMC6690364.

Page 19. Line 369. Was the animal induced before placing on the stereotaxic apparatus?

Author's reply:

Mouse was firstly placed in a chamber lined with paper towel, then a cotton swab saturated with isoflurane was put into the chamber. Once the mouse has lost the righting reflex and breathed in a deeper and slower pattern, the mouse was transferred to stereotaxic apparatus and placed with respiratory mask.

Page 19. Line 371. Tuned → turned

Author's reply:

The corresponding sentence was revised.

Line 426 in revised manuscript:

...turned down to 0.5 % isoflurane during electrophysiology recording...

Page 19. Line 371. I note that anesthesia was used at 1.5% then reduced to 0.5%. But it seems as if a surgical plane of anesthesia can't be maintained at this level. To what extent could the animal have been conscious, without a period of recovery? Were any analgesics employed, such as lidocaine, around the craniotomy?

Author's reply:

Thanks for your comment. We didn't check to what extent the mice could have been conscious. Under 0.5 % isoflurane, the mice were in anesthesia and showed no responses when touched. No analgesics were applied during the experiments.

Page 19. Line 380. Throughout whole recordings.

Author's reply:

The corresponding sentence was revised.

Line 433-435 in revised manuscript:

The craniotomy was filled with fresh warm sterile buffered saline (150 mM NaCl, 2.5 mM KCl, 10 mM HEPES, pH 7.4) throughout the entire recordings.

Reviewers' Comments:

Reviewer #1:

Remarks to the Author:

The authors have addressed all the reviewers' comments accordingly. I agree with most of the additional results, except the patch clamp results (Fig. S19).

It seems that the power density required for cell activation is too low ($\sim 0.001 \text{ mW/mm}^2$). These results are contradictory with most threshold intensity reported in the literature, which is usually $\sim 1 \text{ mW/mm}^2$. The difference of three orders of magnitude seems unreasonable. One can list numerous reports on this.

For example,

1. ChR2

Table 1. Channel and kinetic properties of ChR2, ChR2/H134R, ChETA, VChR1, ChD, ChEF and ChIEF

Channel variant	Response spectra peak		Level of desensitization	Light sensitivity/ EC_{50}		Opening rate τ (ms)	Closing rate τ (ms)
	Peak response	Steady-state response	$I_{\text{steady-state}}/I_{\text{peak}}$	Peak response	Steady-state response	19.8 mW mm^{-2} light intensity	
ChR2	$\sim 470 \text{ nm}$	$\sim 450 \text{ nm}$	~ 0.22 (470 nm)	$\sim 1.10 \text{ mW mm}^{-2}$	$\sim 1.05 \text{ mW mm}^{-2}$	$\sim 1.21 \text{ ms}$	$\sim 13.5 \text{ ms}$
ChR2/H134R	$\sim 450 \text{ nm}$	$\sim 450 \text{ nm}$	~ 0.39 (470 nm)	$\sim 1.07 \text{ mW mm}^{-2}$	$\sim 0.98 \text{ mW mm}^{-2}$	$\sim 1.92 \text{ ms}$	$\sim 17.9 \text{ ms}$
ChETA	$\sim 490 \text{ nm}^*$		~ 0.24 (470 nm) \ddagger	$\sim 5.02 \text{ mW mm}^{-2}\ddagger$	$\sim 0.62 \text{ mW mm}^{-2}\ddagger$	$\sim 0.86 \text{ ms}\ddagger$	$\sim 7.9\text{--}8.5 \text{ ms}^*\ddagger$
VChR1	$\sim 570 \text{ nm}$	$\sim 550 \text{ nm}$	~ 0.48 (570 nm)	Not tested	Not tested	$\sim 2.8 \text{ ms}$	$>90 \text{ ms}\ddagger$
						(15 mW mm^{-2+})	
ChD	$\sim 450 \text{ nm}$	$\sim 450 \text{ nm}$	~ 0.31 (470 nm)	$\sim 3.23 \text{ mW mm}^{-2}$	$\sim 1.02 \text{ mW mm}^{-2}$	$\sim 1.49 \text{ ms}$	$\sim 7.82 \text{ ms}$
ChEF	$\sim 470 \text{ nm}$	$\sim 490 \text{ nm}$	~ 0.70 (470 nm)	$\sim 0.72 \text{ mW mm}^{-2}$	$\sim 0.46 \text{ mW mm}^{-2}$	$\sim 1.56 \text{ ms}$	$\sim 24.9 \text{ ms}$
ChIEF	$\sim 450 \text{ nm}$	$\sim 450 \text{ nm}$	~ 0.80 (470 nm)	$\sim 1.65 \text{ mW mm}^{-2}$	$\sim 1.38 \text{ mW mm}^{-2}$	$\sim 1.62 \text{ ms}$	$\sim 12.0 \text{ ms}$

Definition: $I_{\text{steady-state}}/I_{\text{peak}}$, steady-state current divided by peak current. Most values are from Lin *et al.* (2009a,b), except for * from Gunaydin *et al.* (2010), \ddagger from Zhang *et al.* (2008) and \ddagger J.Y. Lin, unpublished results. Modified with permission from Lin *et al.* (2009b).

From:

<https://physoc.onlinelibrary.wiley.com/doi/pdf/10.1113/expphysiol.2009.051961>

2. CIV1

From:

<https://www.nature.com/articles/nmeth.1808.pdf>

3. ChrimsonR

From:

<https://www.nature.com/articles/nmeth.2836.pdf>

One can hardly believe that a power density as low as 0.001 mW/mm² can be applied for optogenetic activation.

I suggest the authors either find references to support their claims, or carefully calibrate their light sources again.

Reviewer #2:

Remarks to the Author:

The authors add some new experiments to improve the manuscript. But not all of the comments are fully addressed. Several issues need further explanation and clarification. Publications in top-tier journals like nature communications should address technical questions and enlighten readers.

1. The prospect of the technique is not clear. To avoid overheating issues, laser power and laser pulse frequency must be kept low (reply to question #11-12 of reviewer 1). The adverse consequences are weak light delivery and low chromatic selectivity (reply to question #7 of reviewer 2). If no feasible solutions to this problem are available, the research may shrink in future.

2. Trichromatic upconversion emission has been addressed before. So the innovation of this report is tri-chromatic optogenetic modulation. The animal experiments, however, were limited to only two colours (reviewer 3 also has this concern). Were there any technical difficulties using three colours, or are three colours just not needed in practical applications?

3. Reply to question #4.2 of reviewer 2, the authors stated that the core's red emission suppression was due to the de-saturation of Yb. But how did this de-saturation affect the shell's filtration/blocking? A mathematical model and fitting of the data (Er intensity against Tm concentration) would help.

4. Reply to question #5 of reviewer 2, the absorbance of different shells is compared. However, how much light intensity was left after the excitation light passes through the shell was not given. A filtration/blocking effect should result in very low transmitted light intensity.

Reviewer #3:

Remarks to the Author:

The authors have provided an extensive revision of the manuscript. In my opinion, they have now responded adequately to my concerns. The addition of the new biological data is welcome and has added to this document.

Point-to-Point response to reviewers' comments:

Reviewer #1 (Remarks to the Author):

The authors have addressed all the reviewers' comments accordingly. I agree with most of the additional results, except the patch clamp results (Fig. S19). It seems that the power density required for cell activation is too low ($\sim 0.001 \text{ mW/mm}^2$). These results are contradictory with most threshold intensity reported in the literature, which is usually $\sim 1 \text{ mW/mm}^2$. The difference of three orders of magnitude seems unreasonable. One can list numerous reports on this.

For example,

1. ChR2

Table 1. Channel and kinetic properties of ChR2, ChR2/H134R, ChETA, VChR1, ChD, ChEF and ChIEF

Channel variant	Response spectra peak		Level of desensitization	Light sensitivity/ EC_{50}		Opening rate τ (ms)	Closing rate τ (ms)
	Peak response	Steady-state response	$I_{\text{steady-state}}/I_{\text{peak}}$	Peak response	Steady-state response	19.8 mW mm^{-2} light intensity	
ChR2	$\sim 470 \text{ nm}$	$\sim 450 \text{ nm}$	~ 0.22 (470 nm)	$\sim 1.10 \text{ mW mm}^{-2}$	$\sim 1.05 \text{ mW mm}^{-2}$	$\sim 1.21 \text{ ms}$	$\sim 13.5 \text{ ms}$
ChR2/H134R	$\sim 450 \text{ nm}$	$\sim 450 \text{ nm}$	~ 0.39 (470 nm)	$\sim 1.07 \text{ mW mm}^{-2}$	$\sim 0.98 \text{ mW mm}^{-2}$	$\sim 1.92 \text{ ms}$	$\sim 17.9 \text{ ms}$
ChETA	$\sim 490 \text{ nm}^*$		~ 0.24 (470 nm) \ddagger	$\sim 5.02 \text{ mW mm}^{-2}\ddagger$	$\sim 0.62 \text{ mW mm}^{-2}\ddagger$	$\sim 0.86 \text{ ms}\ddagger$	$\sim 7.9\text{--}8.5 \text{ ms}\ddagger$
VChR1	$\sim 570 \text{ nm}$	$\sim 550 \text{ nm}$	~ 0.48 (570 nm)	Not tested	Not tested	$\sim 2.8 \text{ ms}$ (15 mW mm^{-2})	$>90 \text{ ms}$
ChD	$\sim 450 \text{ nm}$	$\sim 450 \text{ nm}$	~ 0.31 (470 nm)	$\sim 3.23 \text{ mW mm}^{-2}$	$\sim 1.02 \text{ mW mm}^{-2}$	$\sim 1.49 \text{ ms}$	$\sim 7.82 \text{ ms}$
ChEF	$\sim 470 \text{ nm}$	$\sim 490 \text{ nm}$	~ 0.70 (470 nm)	$\sim 0.72 \text{ mW mm}^{-2}$	$\sim 0.46 \text{ mW mm}^{-2}$	$\sim 1.56 \text{ ms}$	$\sim 24.9 \text{ ms}$
ChIEF	$\sim 450 \text{ nm}$	$\sim 450 \text{ nm}$	~ 0.80 (470 nm)	$\sim 1.65 \text{ mW mm}^{-2}$	$\sim 1.38 \text{ mW mm}^{-2}$	$\sim 1.62 \text{ ms}$	$\sim 12.0 \text{ ms}$

Definition: $I_{\text{steady-state}}/I_{\text{peak}}$, steady-state current divided by peak current. Most values are from Lin et al. (2009a,b), except for * from Gunaydin et al. (2010), \ddagger from Zhang et al. (2008) and \ddagger J.Y. Lin, unpublished results. Modified with permission from Lin et al. (2009b).

From:

<https://physoc.onlinelibrary.wiley.com/doi/pdf/10.1113/expphysiol.2009.051961>

2. C1V1

From:

<https://www.nature.com/articles/nmeth.1808.pdf>

3. *ChrimsonR*

From:

<https://www.nature.com/articles/nmeth.2836.pdf>

One can hardly believe that a power density as low as 0.001 mW/mm² can be applied for optogenetic activation.

I suggest the authors either find references to support their claims, or carefully calibrate their light sources again.

Author's reply:

Thanks a lot for your comment. First, we calibrated our light sources carefully as the reviewer suggested. In our previous experiments, we measured the light power using optical power meter (Thorlabs, PM100D) and the diameter of the light spot by manually

measuring the light illuminated area on a piece of paper placed underneath the 40X water-immersion objective without careful calibration of the focal plane, which may result in a bigger diameter than the actual one. The light intensity was calculated as the light power divided by the illuminated area, which may lead to a small light intensity. We re-measured the diameter of the illuminated area at the focal plane of the 40X objective through the digital camera to obtain calibrated light intensities, which is in the range of 0.01-0.04 mW/mm² (Figure 1 a-b in this reply letter).

We thank the reviewer for citing relevant references (Ref 1-3). We noted that the pulse durations used in Ref 1-3 are in the range of μ sec – msec, much shorter than that we used in our study. For the measurement of the spectral responses of ChR2, light source was provided with on/off times of 10 μ sec pulses (Ref 1). For C1V1, the same measurement was used (Ref 2), i.e., pulse stimulation with on/off times of 10 μ sec. For ChrimsonR, the laser pulses are 5 msec duration, 5 Hz (Ref 3). To investigate whether short pulse duration results in large stimulation power, we conducted brain slice experiments using 5-msec pulses (as opposed to 1-sec pulses in our previous manuscript) and found that the activation intensity for ChR2-expressing cells was 0.02 mW/mm² at 470 nm (Figure 1c in this reply letter).

We noticed that in Ref 3 that the reviewer cited, 0.015 mW/mm² light at 617 nm could induce proboscis extension reflex (PER) in Chrimson-expressing adult flies (Ref 3, Figure 1d in this reply letter). Meanwhile, in *Curr Biol* 16, 1741-1747 (2006) (Ref 4), ~0.02 mW/mm² light at 470 nm could induce saturated contraction in *Drosophila* larvae expressing ChR2 (Ref 4, Figure 1e in this reply letter). In *Neuroscience* 449, 165-188(2020) (Ref 5), spiking sequence around 40 Hz has been generated using 0.02 mW/mm² light at 595 nm and inhibited by 0.04 mW/mm² light at 490 nm in Chrimson-GtACR2-expressing hippocampal neurons, and the spiking sequence could also be generated using 2.5 mW/mm² light at 595 nm and be inhibited by 0.015 mW/mm² light at 490 nm in vf-Chrimson-GtACR2-expressing hippocampal neurons (Ref 5, Figure 1f in this reply letter).

Figure 1 in this reply letter. Visible light excitation of neural populations. (a) Left, normalized responses of C1V1(T/T)-expressing cells ($n = 6$) and ChR2-expressing cells to 470 nm emission ($n = 3$). Right, example of ChR2-(C1V1)-expressing cell to 470 nm emission. (b) Left, normalized responses of C1V1-expressing cells ($n = 9$) and ChrimsonR-expressing cells to 546 nm emission ($n = 5$). Right, example of C1V1-(ChrimsonR-) expressing cell to 546 nm emission. (c) Example responses of 470 nm activation of ChR2-expressing cell with a 5-msec light pulse duration. (d) Proboscis extension reflex (PER) of flies (pUAS-Chrimson-mVenus in attP18/w-;Gr64f-Gal4/+;Gr64f-Gal4/+, shown as Gr64f x Chrimson) to 25 pulses of lights at 470 nm, 617 nm, 720 nm (cited from Ref 3). (e) Left, wavelength dependency of contractions in larvae expressing channelrhodopsin-2 panneuronally (elav-Gal4/UAS:ChR2) ($n = 10$). Right, Light intensity dependence of contractions in larvae expressing channelrhodopsin-2 panneuronally ($n = 20$), presented as a function of irradiance. Cited from Ref 4. (f) Upper, in Chrimson-GtACR2-expressing neurons, spikes at 40 Hz have been evoked using continuous light pulse of 2 sec at $I_{595} = 0.02 \text{ mW/mm}^2$, and inhibited using 1 sec light pulse at $I_{490} = 0.04 \text{ mW/mm}^2$. Lower, in vf-Chrimson-GtACR2-expressing neurons, spikes at 40 Hz have been evoked using continuous light pulse of 2 sec at $I_{595} = 2.5 \text{ mW/mm}^2$, and inhibited using 1 sec light pulse at $I_{490} = 0.015 \text{ mW/mm}^2$. Cited from Ref 5.

Reference:

- [1] Lin J Y, Lin M Z, Steinbach P, et al. Characterization of Engineered Channelrhodopsin Variants with Improved Properties and Kinetics. *Biophysical Journal*, 2009, 96(5):1803-1814.
- [2] Mattis J, Tye K M, Ferenczi E A, et al. Principles for applying optogenetic tools derived from direct comparative analysis of microbial opsins. *Nat Methods* 9, 159–172 (2012).
- [3] Klapoetke N C. et al. Independent optical excitation of distinct neural populations. *Nat Methods*, 2014, 11(3): 338-46. doi: 10.1038/nmeth.2836.
- [4] Schroll C. et al. Light-induced activation of distinct modulatory neurons triggers appetitive or aversive learning in *Drosophila* larvae. *Curr Biol*. 2006 Sep 5;16(17):1741-7. doi: 10.1016/j.cub.2006.07.023.
- [5] Bansal H. et al. Theoretical Analysis of Low-power Bidirectional Optogenetic Control of High-frequency Neural Codes with Single Spike Resolution. *Neuroscience*. 2020 Nov 21;449:165-188. doi: 10.1016/j.neuroscience.2020.09.022.

Reviewer #2 (Remarks to the Author):

The authors add some new experiments to improve the manuscript. But not all of the comments are fully addressed. Several issues need further explanation and clarification. Publications in top-tier journals like nature communications should address technical questions and enlighten readers.

1. The prospect of the technique is not clear. To avoid overheating issues, laser power and laser pulse frequency must be kept low (reply to question #11-12 of reviewer 1). The adverse consequences are weak light delivery and low chromatic selectivity (reply to question #7 of reviewer 2). If no feasible solutions to this problem are available, the research may shrink in future.

Author's reply:

Thanks a lot for your comment. We agree with the reviewer that due to the combination of overheating issue as well as the limitation in the response threshold, which existed for other NIR studies, the laser power needs to be kept at a defined range, as indicated in Fig. 4f in the revised manuscript (Figure 2 in this reply letter). Though we provided stimulation conditions that meet the requirements in this manuscript, the temperature rise needs to be measured in advance when applying this technique to other studies. Despite limitations in the present work, which need further studied, we thought that the parameters and conditions explored in this paper could be an inspiration and reference for the community. We have added more discussions about the limitations and future prospects in revised manuscript.

(1) Regarding overheating issue, we further explored the upper limit for laser power intensity under different stimulating patterns in order to show how to mitigate thermal effect in NIR optogenetics. As shown in *reply to question # 11 of reviewer 1*, for tri-wavelength stimulation, the temperature increase is less than 2 °C when the pulse duration is 0.1 sec at 5 Hz for a total duration of 120 sec (0.8 mW/mm² at 808 nm, 1.27 mW/mm² at 980 nm, and 1.27 mW/mm² at 1532 nm NIR light, 120 sec duration, Figure 3a in this reply letter). For dual-wavelength stimulation without 1532-nm in behavior experiments (Fig. 5 in the revised manuscript, 59.1 mW/mm² at 808 nm, 38.8 mW/mm² at 980 nm, 120 sec duration), we re-measured the temperature rise, which is about 4.5 °C (Figure 3a in this reply letter). Note that very few optogenetic studies require prolonged stimulation for over 100 sec, we further measured the temperature rises using shorter pulses (50 msec or 25 msec) at higher stimulation frequencies (10 Hz or 20 Hz) for a total duration of 10 sec or 20 sec (Figure 3b in this reply letter). A summary of power densities of three NIR lasers, frequency, total duration, and corresponding temperature rise are shown in Figure 3c in this reply letter. These results showed that to keep the temperature increase to be below 2 °C, decreasing the pulse duration (to 25 - 50 msec) and increasing stimulation frequency (to 10 - 20 Hz) resulted in higher power density. As a result, the upper boundary for light intensity can be extended by 87.5% at 808 nm, 116.5% at 980 nm and 25.2% at 1532 nm. We've added these results into the revised manuscript. In addition, the upconversion efficiency of UCNPs could be further increased, e.g. by thickening the shell layer.

Figure 2 in this reply letter. Figure 4f in the revised manuscript. Calculated allowable range of NIR power density at brain surface. The operational power density is allowed for chromatic selectivity of optogenetic proteins and thermal limit as well as detecting the

activation response. Note that the operational NIR power densities are used in the *in vivo* electrophysiology experiment.

Line 419 – 424 in revised manuscript:

Limitations of the study

On the one hand, due to the combination of overheating issue as well as the limitation in the response threshold, which also existed for other NIR studies, the laser power needs to be kept at a restricted range. We explored the stimulation conditions that meet the requirements in this study, which suggest decreasing pulse duration (25 - 50 msec) and increasing stimulation frequency (10 - 20 Hz) will allow higher power density considering the NIR heating effect. Nevertheless, when applying this technique to other studies, the parameters still need specific calibration in order to control temperature rise.

Figure 3 in this reply letter. Supplementary Figure 30 in the revised manuscript. Curves of temperature rises under different conditions. (a) Purple line, the power densities of 808 nm, 980 nm and 1532 nm laser are 0.8 mW/mm², 1.27 mW/mm² and 1.27 mW/mm², respectively. Orange line, the power densities of 808 nm, 980 nm are 59.1 mW/mm², 38.8 mW/mm², respectively. Stimulation pattern: 5 Hz, 0.1 sec on, 0.1 sec off, 120 sec in total. The red bar indicates time period of irradiation and the left and right gray

column indicates the starting and ending frames of irradiation. (b) Blue curve, 0.8 mW/mm² 808 nm, 3.74 mW/mm² 980 nm and 1.43 mW/mm² 1532 nm laser, 50 msec on, 50 msec off, total duration 20 sec. Purple curve, 1.5 mW/mm² 808 nm, 2.75 mW/mm² 980 nm and 1.59 mW/mm² 1532 nm laser, 25 msec on, 25 msec off, total duration 10 sec. Yellow curve, 0.8 mW/mm² 808 nm, 2.75 mW/mm² 980 nm and 1.59 mW/mm² 1532 nm laser, 25 msec on, 25 msec off, total duration 20 sec. (c) Summary of used conditions and corresponding temperature rise.

Line 328 - 333 in revised manuscript:

Very few optogenetic studies require prolonged stimulation for over 100 s. So we further explored the feasible conditions meeting the requirements of temperature rise (below 2 °C) with higher power when the pulse frequency is 10 Hz or 20 Hz for a shorter total duration of 10 s or 20 s . By decreasing the pulse duration to 25 - 50 ms and increasing the stimulation frequency to 10 - 20 Hz, the upper boundary for light intensity is increased by 87.5% at 808 nm and 116.5% at 980 nm and 25.2% at 1532 nm. (Supplementary Fig. 30b-c).

(2) Regarding the chromatic selectivity, in *reply to question #7 of reviewer 2* (Fig. 22c-d in the previous reply letter, Figure 4c-d in this reply letter), it is shown that the chromatic selectivity is not dependent on stimulation power density above the activation threshold. An effective improvement to the chromatic selectivity is to use opsins with less spectral overlap in future studies. For example, the HfACR1 is one of the most red-shifted channelrhodopsins known, published in PNAS, 2020 (Ref 6). The response ratio of HfACR1 for 475 nm, 540 nm and 650 nm is about 2 : 5 : 5, while that of Chrimson is about 3 : 7 : 3 (Figure 5 in this reply letter). We believe that new red-shifted channelrhodopsins with better spectral segregation would further improve the multi-chromatic optogenetic neural circuit dissection in the near future.

Figure 4 in this reply letter. Figure 22 in previous reply letter. Supplementary Figures 18 and 28 in the revised manuscript. Comparison of green/red emission ratio under 808 nm excitation with C1V1(T/T) and ChrimsonR response ratio. (a) Photos of UCNPs emission power measurement set up. **(b)** Table of measured visible light power under the same laser output to in vivo experiments. **(c)** Recorded cellular level neuronal activities by patch clamp from brain slices expressed C1V1(T/T) and ChrimsonR respectively under different visible light powers ($n = 9$ for C1V1(T/T) expressing-cells and $n = 5$ for ChrimsonR expressing-cells). **(d)** In vivo normalized responses of neurons with 530 and 650 nm laser under different power intensities.

Figure 5 in this reply letter. (a) Cited from *question # 1 of reviewer1* in previous reply letter. Responses of Chrimson for 475 nm, 540 nm and 650 nm light, indicated by red curve. (b) Cited form Ref 6. Responses of HfACR1 for 475 nm, 540 nm and 650 nm light.

Reference:

[6] Govorunova E G , Sineshchekov O A , Li H , et al. RubyACRs, nonalgal anion channelrhodopsins with highly red-shifted absorption. *Proceedings of the National Academy of Sciences*, 2020, 117(37):202005981.

Line 424 – 429 in revised manuscript:

On the other hand, although we can address the spectral overlap between ChrimsonR and Chr2/C1V1 by finding the combination of light intensities, which gives an independent activation of different optogenetic proteins in our experiments. Using opsins with less spectral overlap would be a more effective improvement. For example, the newly reported HfACR1 is red-shifted farther than ChrimsonR³⁹. We believe that new red-shifted channelrhodopsins with better spectral segregation would further improve the multi-chromatic optogenetic neural circuit dissection in the near future.

2. Trichromatic upconversion emission has been addressed before. So the innovation of this report is tri-chromatic optogenetic modulation. The animal experiments, however, were limited to only two colours (reviewer 3 also has this concern). Were there any technical difficulties using three colours, or are three colours just not needed in practical applications?

Author's reply:

We thank the reviewer for the comment. Tri-chromatic optogenetic modulation (one dimensional corridor locomotion test) was carried out with the results presented in the

previous version of revised manuscript as Supplementary Figure 31 (Figure 6 in this reply letter). In the tri-chromatic modulated locomotion test, 808 nm or 1532 nm stimulation induced increased travelled distance while 980 nm stimulation caused decreased travelled distance. When 808 nm, 980 nm and 1532 nm lasers were all illuminated onto the skull, average distance traveled was smaller than that in control mice.

As for the potential application of trichromatic optogenetics, it is actually highly demanded in neuroscience. Because information was processed within the brain network multi-directionally. The more measurable inputs with varying levels of coherence from multiple regions will allow the study of activities contribution to the integration of sensory information. Studying the brain circuits with one or two optogenetic protein would only allow unidirectional dissection of the behavior.

Figure 6 in this reply letter, Supplementary Figure 31 in revised manuscript. Transcranial optogenetic multi-color manipulation of mouse locomotion behavior using UCNPs. (a) Running speed of example trials in one-dimensional corridor in VR system under 0.8 mW/mm^2 808 nm, 1.27 mW/mm^2 980 nm, 1.27 mW/mm^2 1532 nm and trichromatic NIR excitations. **(b)** Delta running distances in one-dimensional corridor in VR system under 0.8 mW/mm^2 808 nm, 1.27 mW/mm^2 980 nm, 1.27 mW/mm^2 1532-nm and trichromatic NIR excitations compared with control (no light stimulation) trials. ($n = 4$ mice, 3 trials for each mouse).

3. Reply to question #4.2 of reviewer 2, the authors stated that the core's red emission suppression was due to the de-saturation of Yb. But how did this de-saturation affect the shell's filtration/blocking? A mathematical model and fitting of the data (Er intensity against Tm concentration) would help.

Author's reply:

Thanks for your comment, a precisely mathematical model with data-fitting is actually unavailable in a simplified qualitative model that used for analyzing the energy distribution. We add more interpretation to explain how the de-saturation of Yb affects the inner emission intensity in the revising manuscript.

In the excitation of core-shell UCNPs, the unconverted photons will penetrate the outer layers and get into the inner core. The emission of Er^{3+} is positively correlative with the incident photons. The rate of photons that entering the core is:

$$w = w_0 - w_1$$

where w_0 is the rate of total incident photons and w_1 is the rate of converted photons in outer layer. In the outer layers, the energy absorbing rate is σP , where σ is the absorption cross-section and P is the excitation power density. But only sensitizers in ground state of outer layer are able to convert the photons, so that the photon converting rate of sensitizer in outer layer is:

$$w_1 = \frac{\sigma P}{h\nu} N_g = \frac{\sigma P}{h\nu} (N - N_a)$$

where N_g , N_a are the ground state and active state population with the constant total population $N = N_g + N_a$. This equation means that lowering the outer layer sensitizer activated population N_a can increase the converting rate w_1 under the same excitation power and subsequently reducing the photons approaching the inner core (w).

Tm^{3+} , as the energy consumer, is able to extract energy from Yb^{3+} in the activated state allowing Yb^{3+} returns to ground state. This is the “de-saturation effect” of Yb^{3+} . The activated population of sensitizer discussed after supplementary information Fig.6 in previous manuscript is:

$$N_a = \frac{\frac{\sigma P}{h\nu} N}{\sum_i W_i N_i + R_a + \frac{\sigma P}{h\nu}} = \frac{P}{A + B \cdot P}$$

when increasing the excitation power P , the activated state population N_a will approach to the total population N . Doping of Tm delays this saturation process by adding extra energy transfer routes in the coefficient A . This doping gives lower N_a under the same excitation power density (Figure 7 in this reply letter) thus reducing w as discussed above.

Figure 7 in this reply letter. Simulation curves of sensitizer activated state population as a function of excitation power density with different Tm doping concentration.

Line 549 – 566 in revised supplementary information:

In the excitation of core-shell UCNPs, the unconverted photons will penetrate the outer layers and get into the inner core. The emission of Er^{3+} is positively correlative with the incident photons. The rate of photons that entering the core is:

$$w = w_0 - w_1$$

where w_0 is the rate of incident photons and w_1 is the rate of converted photons in outer layer. The energy absorbing rate is σP , where σ is the absorption cross-section and P is the excitation power density. But only sensitizers in ground state of outer layer are able to convert the photons, so that the photon converting rate of sensitizer in outer layer is:

$$w_1 = \frac{\sigma P}{h\nu} N_g = \frac{\sigma P}{h\nu} (N - N_a)$$

It means that lowering the outer layer sensitizer activated population N_a can increase the converting rate w_1 under the same excitation power and subsequently reducing the photons approaching the inner core (w).

Tm^{3+} , as the energy consumer, is able to extract energy from Yb^{3+} in the activated state allowing Yb^{3+} returns to ground state. When increasing the excitation power P , the activated state population N_a will increase to the total population N . Doping of Tm delays this saturation process by adding extra energy transfer routes. Consequently, at a specified excitation power density P , the population density of active state N_a is decreased with the increasing coefficient A thus reducing the rate of photons approaching to inner core w .

4. Reply to question #5 of reviewer 2, the absorbance of different shells is compared. However, how much light intensity was left after the excitation light passes through the shell was not given. A filtration/blocking effect should result in very low transmitted light intensity.

Author's reply:

Actually, the light intensity after shells penetration is inaccessible by directly experimental measuring. Instead, we deduced the transmitted ratio from the change of inner emission intensity. The sensitized upconversion emission intensity is:

$$I = k \cdot P^n$$

where I is the upconversion emission intensity, P is the excitation power density, n is the number of sensitizing steps and k is a constant coefficient. So that the relationship of excitation power and inner emission intensity ratio is:

$$\frac{I_1}{I_0} = \left(\frac{P_1}{P_0}\right)^n$$

where I_1, I_0 are inner emission intensity with or without dissipation shell, P_1, P_0 are the excitation power density with or without dissipation shell. For ~651 nm red emission band from Er, the number of sensitizing steps n is ~2 (CR involved). From experimental results, a shell structure of 49%Yb1%Tm will gives 56.4% emission suppression (without considering Tm^{3+} emission contribution), which suggest less than 66% (P_1/P_0) of excitation intensity was left after penetration. Similarly, a structure of 80%Yb1%Tm gives less than 60% intensity left. That is to say for obtaining an effective decreasing of inner emission, an extremely low transmitted excitation light intensity may not be necessary, especially for higher multi-photon process. Because the emission intensity has an exponential dependency to excitation power density, which suggests that excitation decreasing will leads to exponential drop of emission. Considering only absorption process may lead to improper intuition as for inner emission suppression. These discussions are added in revised supplementary information.

Line 407 – 414 in revised supplementary information:

We can deduce the transmitted ratio from the change of inner emission intensity with or without dissipation shell. The relationship of excitation power and inner emission intensity ratio is:

$$\frac{I_1}{I_0} = \left(\frac{P_1}{P_0}\right)^n$$

where I_1, I_0 are inner emission intensity with or without dissipation shell, P_1, P_0 are the excitation power density with or without dissipation shell. For ~651 nm red emission band from Er, the number of sensitizing steps n is ~2 (CR involved). From experimental results, a shell structure of 49%Yb1%Tm will give 56.4% emission suppression (without

considering Tm^{3+} emission contribution), which means less than 66% (P_1/P_0) of excitation intensity was left after penetration. Similarly, a structure of 80%Yb1%Tm gives less than 60% intensity left.

Reviewer #3 (Remarks to the Author):

The authors have provided an extensive revision of the manuscript. In my opinion, they have now responded adequately to my concerns. The addition of the new biological data is welcome and has added to this document.

Reviewers' Comments:

Reviewer #1:

Remarks to the Author:

The authors have responded to all the reviewers' comments properly and revised the manuscript accordingly. I suggest the paper can be accepted.

Reviewer #2:

Remarks to the Author:

The authors have tried their best to address the review comments. I have no other questions.

Reviewer #1 (Remarks to the Author):

-The authors have responded to all the reviewers' comments properly and revised the manuscript accordingly. I suggest the paper can be accepted.

Author's reply:

We're appreciate for the constructive advice from the reviewers.

Reviewer #2 (Remarks to the Author):

-The authors have tried their best to address the review comments. I have no other questions.

Author's reply:

We're appreciate for the constructive advice from the reviewers.